# A data-driven approach for timescale decomposition of biochemical reaction networks

Amir Akbari,[1] Zachary B. Haiman,[1] Bernhard O. Palsson[1,2]

**ABSTRACT** Understanding the dynamics of biological systems in evolving environments is a challenge due to their scale and complexity. Here, we present a computational framework for the timescale decomposition of biochemical reaction networks to distill essential patterns from their intricate dynamics. This approach identifies timescale hierarchies, concentration pools, and coherent structures from time-series data, providing a system-level description of reaction networks at physiologically important timescales. We apply this technique to kinetic models of hypothetical and biological pathways, validating it by reproducing analytically characterized or previously known concentration pools of these pathways. Moreover, by analyzing the timescale hierarchy of the glycolytic pathway, we elucidate the connections between the stoichiometric and dissipative structures of reaction networks and the temporal organization of coherent structures. Specifically, we show that glycolysis is a cofactor-driven pathway, the slowest dynamics of which are described by a balance between high-energy phosphate bond and redox trafficking. Overall, this approach provides more biologically interpretable characterizations of network dynamics than large-scale kinetic models, thus facilitating model reduction and personalized medicine applications.

**IMPORTANCE** Complex interactions within interconnected biochemical reaction networks enable cellular responses to a wide range of unpredictable environmental perturbations. Understanding how biological functions arise from these intricate interactions has been a long-standing problem in biology. Here, we introduce a computational approach to dissect complex biological systems' dynamics in evolving environments. This approach characterizes the timescale hierarchies of complex reaction networks, offering a system-level understanding at physiologically relevant timescales. Analyzing various hypothetical and biological pathways, we show how stoichiometric properties shape the way energy is dissipated throughout reaction networks. Notably, we establish that glycolysis operates as a cofactor-driven pathway, where the slowest dynamics are governed by a balance between high-energy phosphate bonds and redox trafficking. This approach enhances our understanding of network dynamics and facilitates the development of reduced-order kinetic models with biologically interpretable components.

**KEYWORDS** kinetic models, timescale decomposition, dynamic-mode decomposition, data-driven approach, coherent structures

Growth and adaptation are hallmarks of all living systems. They are inherently dynamic processes that are orchestrated by interacting networks of biochemical reactions. Understanding the dynamics underpinning the intricate behavior of biological systems has been a major challenge of systems biology. Kinetic models based on detailed enzymatic rate laws can capture the complex interactions and dynamics of

Address correspondence to Bernhard O. Palsson, bpalsson@ucsd.edu.

The authors declare no conflict of interest.

See the funding table on p. 30.

biological systems (1, 2). However, the application of these models to large-scale networks is hindered by numerical challenges and their limited interpretability (3).

Biochemical reaction networks have complex structures. Yet, they often exhibit multi-scale and comparatively simple dynamics due to the presence of timescale hierarchies (4). Separation of timescales is believed to be an essential feature of highly evolved reaction networks, conferring stability and robustness (4, 5). It can lead to the modularization of network dynamics and emergence of "independent" functional units, both of which underlie the robustness and evolvability of living organisms (5, 6). Although our knowledge of the links between biological complexity and functional modularization is limited, we can gain a deeper understanding of the structural evolution and organization of biochemical reaction networks through systematic studies of their timescale hierarchies.

Timescale decomposition is a common approach for analyzing the dynamics of complex systems. Several techniques have been developed for biochemical reaction networks, most of which fall into two main categories: top-down and bottom-up approaches. The first leverages statistical methods and clustering algorithms to identify collections of metabolites with correlated concentration trajectories—referred to as concentration pools—from experimentally measured time-series data (7, 8). These techniques do not rely on evolution equations or kinetic rate laws, although they can use numerically generated time-series data furnished by mass-balance equations. They can handle large-scale systems of varying complexity but do not mechanistically relate concentration pools to the stoichiometric and dissipative structures of reaction networks. The second approach determines timescale hierarchies from steady-state eigenvalues (9, 10). It hinges on precise formulations of all reaction rates, providing a mechanistic association between concentration pools and structural properties of reaction networks. Although computations are tractable for large-scale networks, inaccuracies may arise for networks with highly nonlinear rate laws where Jacobian spectra are time-dependent.

Dimensionality reduction is a key step of timescale decomposition. Several classes of model reduction techniques have been previously developed to reduce the dimensionality of complex biochemical reaction networks (11). Trajectory-based techniques (12) describe the dynamics of the system without relying on any assumptions about timescales, making them well-suited for systems with complex time-dependent patterns. However, these techniques can be computationally intensive and sensitive to noise in the data, requiring substantial amount of data to accurately capture system dynamics (13). Singular perturbation techniques (14, 15) effectively reduce the dimensionality by isolating fast and slow dynamics, making them suitable for systems with disparate timescales. However, they are challenging to automate, requiring the functional form of rate laws and kinetic parameters. Lumping techniques (16, 17) offer simplicity and efficiency in model reduction, but how well the reduced model preserves the dynamic characteristics of the original system depends on an *ad hoc* choice of the lumping function.

In this paper, we develop a computational framework for the timescale decomposition of biochemical reaction networks (Fig. 1), which leverages the strengths of both top-down and bottom-up techniques. This approach, termed dynamic mode analysis (DMA), determines the timescale hierarchy of a reaction network from experimentally measured or numerically generated time-series data. It can identify concentration pools for complex networks with uncharacterized rate laws as reliably as top-down techniques. It can also provide a mechanistic description of concentration pools and their organization with respect to the energetics and stoichiometry of reaction networks in the same way as bottom-up techniques. A key component of DMA is an extension of dynamic mode decomposition (DMD) (18–20)—a technique originally developed to characterize coherent structures arising in fluid flows—that we introduce to identify the dominant exponential decay modes associated with each timescale. We study the timescale hierarchies of hypothetical and biological pathways using DMA, showing that this approach can reproduce the previously characterized concentration pools of these

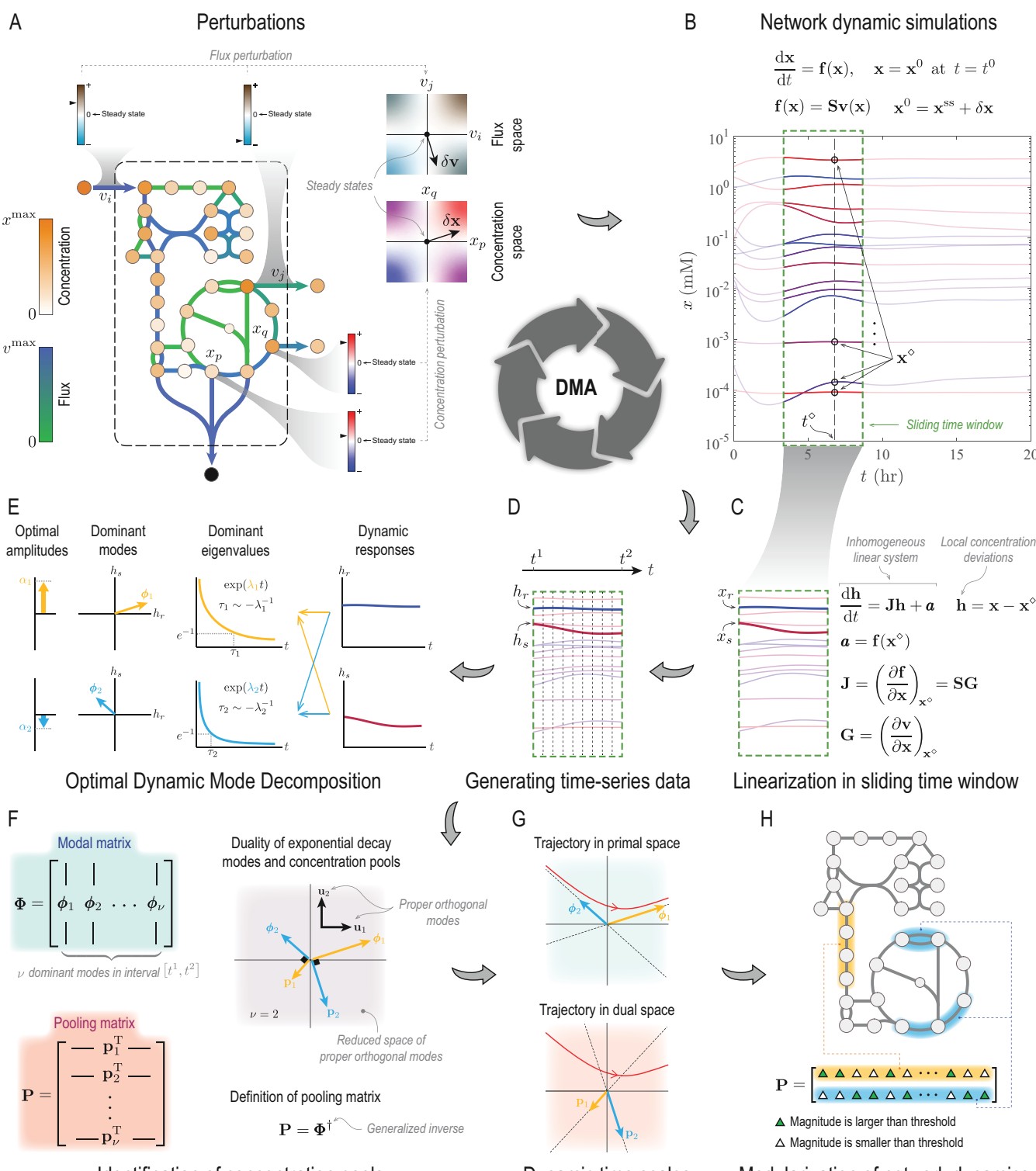

**FIG 1** Overview of DMA. (A) Steady states of biochemical reaction networks are perturbed by introducing concentration or flux disturbances. (B) Concentration trajectories $\mathbf{x}(t)$ are constructed by integrating transient mass-balance equations. (C) For a general nonlinear system, dynamic modes are ascertained by linearizing mass-balance equations in a sliding time window, resulting in a local inhomogeneous linear system. (D) Time-series data are generated by evaluating local concentration deviations $\mathbf{h}$ at $N+1$ equally spaced time points in the interval $[t^1, t^2]$ that spans the time window. (E) An ODMD developed in this work identifies dominant exponential decay modes in the time window from time-series data. (F) The pooling matrix $\mathbf{P}$, defined as the Moore-Penrose inverse (21) of the modal matrix $\mathbf{\Phi}$, is determined from dominant decay modes. (G) Disparate timescales are identified from dominant eigenvalues. (H) Biologically interpretable pools are constructed from the pooling matrix to modularize network dynamics.

pathways. We also establish a connection between the time-delayed autocorrelation matrix—a statistical descriptor used in top-down techniques (8)—and Jacobian spectra, demonstrating why this descriptor is a useful metric for concentration-pool classifiers.

## RESULTS

We illustrate the application of DMA by analyzing the timescale hierarchies of chemical reaction networks in three case studies. The first two are hypothetical pathways that possess key characteristics of typical biological pathways. The last is glycolysis—a well-studied biological pathway, the concentration pools of which were previously characterized (9, 22). We examine this case study to validate our computational framework by reproducing some of the known concentration pools of glycolysis and their respective timescales and to gain insight into its higher-level organization. Three important concepts we encounter in this section are concentration pools, coherent structures, and transitory regimes between two consecutive timescales, which are all defined in Materials and Methods (see "Concentration pools and coherent structures").

### Toy Model 1

The first case study is Toy Model 1, which is the same pathway examined in Materials and Methods (see "Concentration pools and coherent structures" and Fig. S1 and S2). It converts a substrate (metabolite 1) into a product (metabolite 2) without energy coupling. The substrate and product are energetically equivalent, and all the equilibrium constants are of the same order of magnitude ($K_j^{eq} = 1$ for $j = 1 \cdots m$); therefore, the flux is driven through the pathway by maintaining a gradient between the extracellular concentrations $x_1^*$ and $x_4^*$ (Fig. 2A). The boundary reactions (reactions 1 and 5) are the rate-limiting steps, and the rate constants of consecutive intracellular reactions are separated by two orders of magnitude (Fig. S1C).

Toy Model 1 has a timescale hierarchy due to the separation of rate constants. We identified its timescales and their respective concentration pools using DMA (Fig. 2). We induced a dynamic response by perturbing $x_1$ and $x_4$, which activated all three timescales associated with the rate constants of the intracellular reactions (Fig. 2A and B). The successive equilibration of the intracellular flux disturbances partitioned the total relaxation time into seven distinct time intervals, corresponding to the three main timescales and their transitory counterparts we highlighted previously (Fig. 2B and C). Of particular interest are the slowest transients near the steady state. Here, all the intracellular metabolites pool together into a single aggregate metabolite with coefficients $(1, 1, 1, 1)$, the concentration of which is controlled by the boundary reactions. Overall, the timescales and pooling matrices furnished by DMA for each time interval agreed well with our estimates based on the approximate analytical method ("Concentration pools and coherent structures"; Fig. 2C).

Furthermore, DMA could accurately characterize the transitory regime between two consecutive timescales ("Concentration pools and coherent structures"), identifying the disequilibrium and conservation stages associated with the relaxation of each reaction (e.g., see the coefficients of $x_1$ and $x_2$ in the first and second pooling maps of interval 2, characterizing the relaxation of reaction 2). These general characteristics can help understand the dynamic responses of more complex reaction networks.

### Toy Model 2

The second case study, referred to as Toy Model 2, involves the same pathway as in Toy Model 1, but with energy coupling (Fig. S3). It converts a high-energy substrate (metabolite 1) into a low-energy product (metabolite 2). The released energy is then utilized to convert a low-energy cofactor (metabolite 6) into its high-energy counterpart (metabolite 5). The high-energy cofactor drives an uphill step at the beginning (reaction 2) and is recovered in a downhill step at the end (reaction 4). Thus, it serves a similar metabolic function to glycolysis. The cofactors are exchanged with the extracellular

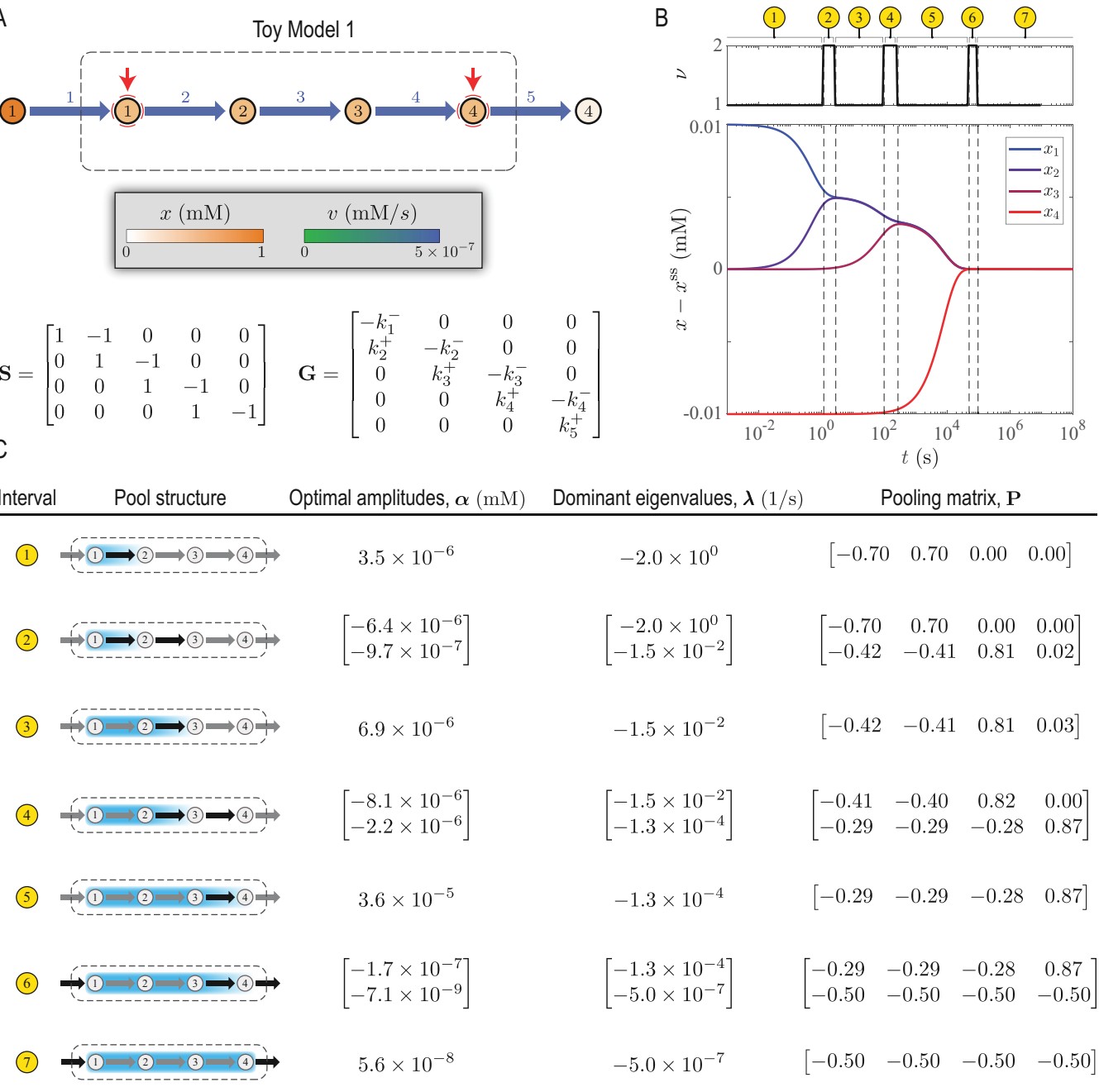

**FIG 2** Dynamic mode analysis of Toy Model 1 described in Fig. S1. (A) Visualization of the steady-state solution in a network map. Red arrows indicate metabolites with perturbed concentrations. (B) Dynamic response to concentration perturbations. Deviations from the steady-state $x - x^{ss}$ and dimensionality $\nu$ are plotted along the dynamic trajectory. Dashed lines indicate a transition between two time periods with distinct exponential decay modes, partitioning the overall relaxation time into seven characteristic time intervals. (C) Optimal amplitudes, dominant eigenvalues, and pooling matrix for each time interval identified in panel B. Black arrows in pool structures indicate parts of the network that are affected the most by the initial perturbation in the corresponding time interval. Gradient and uniformly colored envelopes enclose metabolites, the concentration trajectories of which are negatively and positively correlated in the corresponding time interval, respectively.

environment depending on their concentration gradient across the membrane, and so are the substrate and product. As in Toy Model 1, reactions 1 and 5 are the rate-limiting steps. However, the boundary reactions for the cofactors (reactions 6 and 7) have the largest rates. Unlike Toy Model 1, the mass-balance equations for this model are nonlinear due to the bilinearity of the mass-action rates for the cofactor-coupled

reactions. In this case study, an interplay between the rate-limiting steps, cofactor exchange reactions, and cofactor-coupled intracellular reactions shapes the structure of concentration pools.

We analyzed the timescale hierarchy of Toy Model 2 by examining its dynamic response to concentration perturbations (Fig. 3A). During the relaxation of the ensuing concentration and flux disturbances, DMA identified eight time intervals associated with distinct timescales and concentration pools (Fig. 3B and C). In this model, besides reaction rates, energy coupling also influences the chronology of reactions by linking the equilibration of the cofactor-driven steps. Here, the disequilibrium stages of reactions 2 and 4 occur on the ~1- to 10-s timescale, and their conservation stages occur on the ~10- to 30-s timescale. Once the concentration and flux disturbances of these reactions have relaxed, $x_1$ and $x_3$ form two coherent structures with $x_2$ and $x_4$, respectively, giving rise to two aggregate variables $x_{12} := x_1 + 2x_2$ and $x_{34} := 2x_3 + x_4$. Next, the disequilibrium stage of reaction 3 occurs on the ~100–1,000 s timescale followed by a conservation stage that persists until the steady state has been reached. During this conservation stage, $x_{12}$ and $x_{34}$ pool together, forming a larger coherent structure. Accordingly, the slowest transients are characterized by $x_{1-4}$ pooling together into a single aggregate metabolite with coefficients $(1, 2, 2, 1)$. As the steady state is approached, the concentration of this aggregate metabolite is controlled by reactions 1 and 5.

Finally, we highlight the important role of cofactors in the dynamics of pathways with energy coupling. In this case study, because the cofactor exchange reactions are the fastest in the network, they control the short-term responses ($t \lesssim 30$ s), only influencing the disequilibrium and conservation stages of reactions 2 and 4. Upon concentration perturbation, $x_5$ and $x_6$ become negatively correlated, forming a coherent structure almost immediately ($t \lesssim 1$ s). As mentioned above, the long-term responses of Toy Model 2 are mostly controlled by reactions 1 and 5 near the steady state with minimal effect from the cofactors.

## Glycolysis

Glycolysis is a central energy-conversion pathway in biology (23). It converts a high-energy substrate (glucose) into low-energy products (pyruvate and lactate). The energy released in this process is then used to produce high-energy cofactors (ATP and NADH) (see "Energetics of glycolysis" for details). To generate a sufficient amount of high-energy phosphate bonds, it also imports inorganic phosphate (pi). A key step of glycolysis is catalyzed by fructose-bisphosphate aldolase, splitting fructose 1,6-bisphosphate into the triose phosphates dihydroxyacetone phosphate and glyceraldehyde 3-phosphate. This step makes the energy stored in the chemical bonds of the pentose ring accessible for energy conversion in downstream reactions.

In the third case study, we analyzed the timescale hierarchy of the glycolytic pathway in human red blood cells (Fig. 4). We solved a mass-action kinetic model of glycolysis using the MASSpy package (model parameters are provided in Data S1) (24) and computed its dynamic response to a perturbation in the ATP load (Fig. 4A). Several reactions in upper and lower glycolysis are coupled to ATP hydrolysis, so their relaxations are tightly linked together. These energy-coupling mechanisms impart an autocatalytic structure to the glycolytic pathway, a characteristic of which is oscillatory dynamics (25). Interestingly, during the relaxation of the load perturbation, concentration trajectories exhibit oscillatory dynamics on timescales ranging from minutes to a day, coinciding with the average erythrocyte circulation time and circadian period, respectively (26). We analyzed the concentration trajectories using DMA and identified 11 time intervals with distinct timescales and concentration pools (seven of which are highlighted in Fig. 4B and C). In the following, we highlight a few coherent structures associated with these timescales that were characterized previously. A complete list of the dominant eigenvalues and their respective pooling matrices for all time intervals is provided in Data S2.

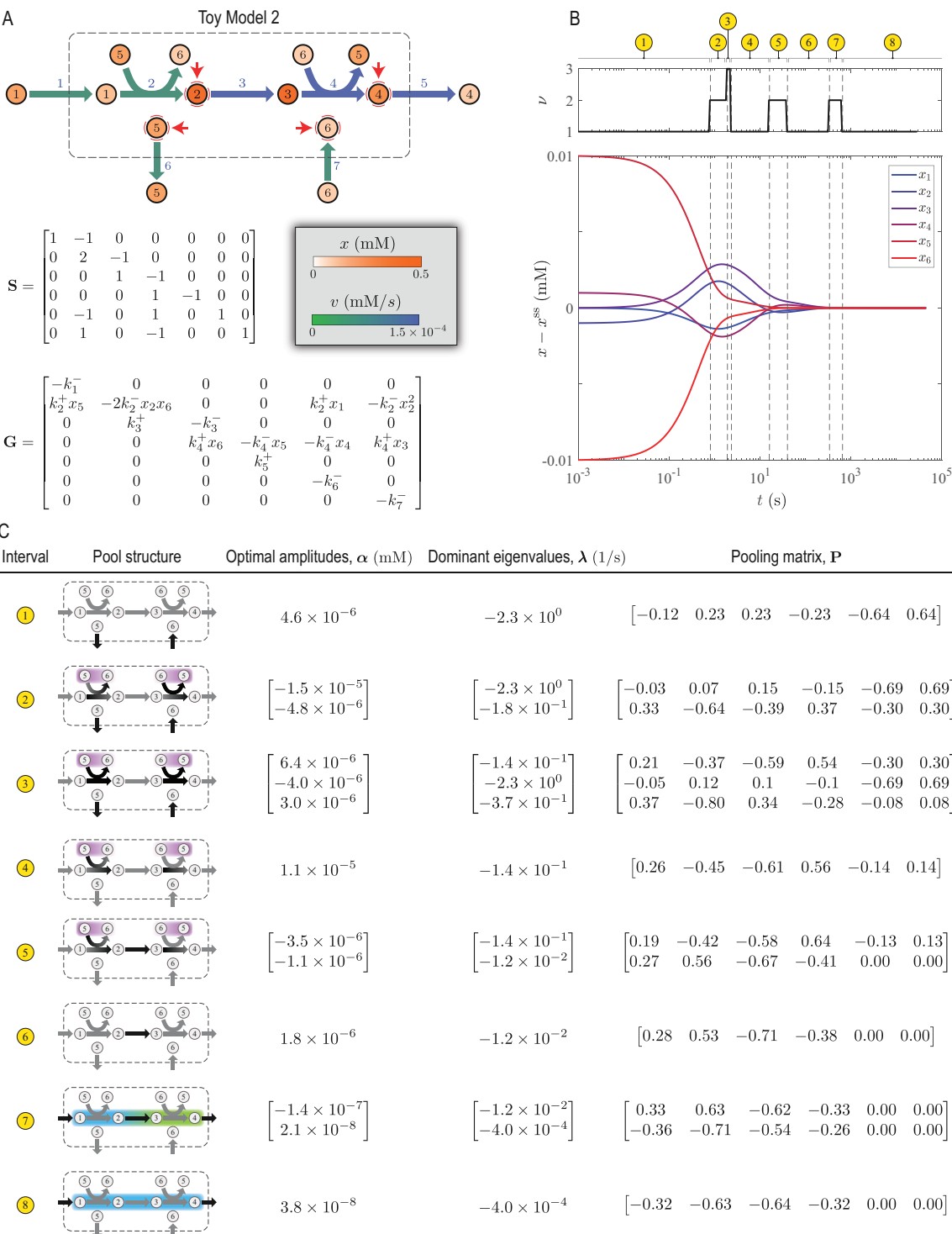

FIG 3  DMA of Toy Model 2 described in Fig. S3. (A) Visualization of the steady-state solution in a network map. Red arrows indicate metabolites with perturbed concentrations. (B) Dynamic response to concentration perturbations. Deviations from the steady-state $x - x^{\text{ss}}$ and dimensionality $\nu$ are plotted along the dynamic trajectory. Dashed lines indicate a transition between two time periods with distinct exponential decay modes, partitioning the overall relaxation time into eight characteristic time intervals. (C) Optimal amplitudes, dominant eigenvalues, and pooling matrix for each time interval identified in panel B. Black arrows in pool structures indicate parts of the network that are affected the most by the initial perturbation in the corresponding time interval. Gradient and uniformly colored envelopes enclose metabolites, the concentration trajectories of which are negatively and positively correlated in the corresponding time interval, respectively.

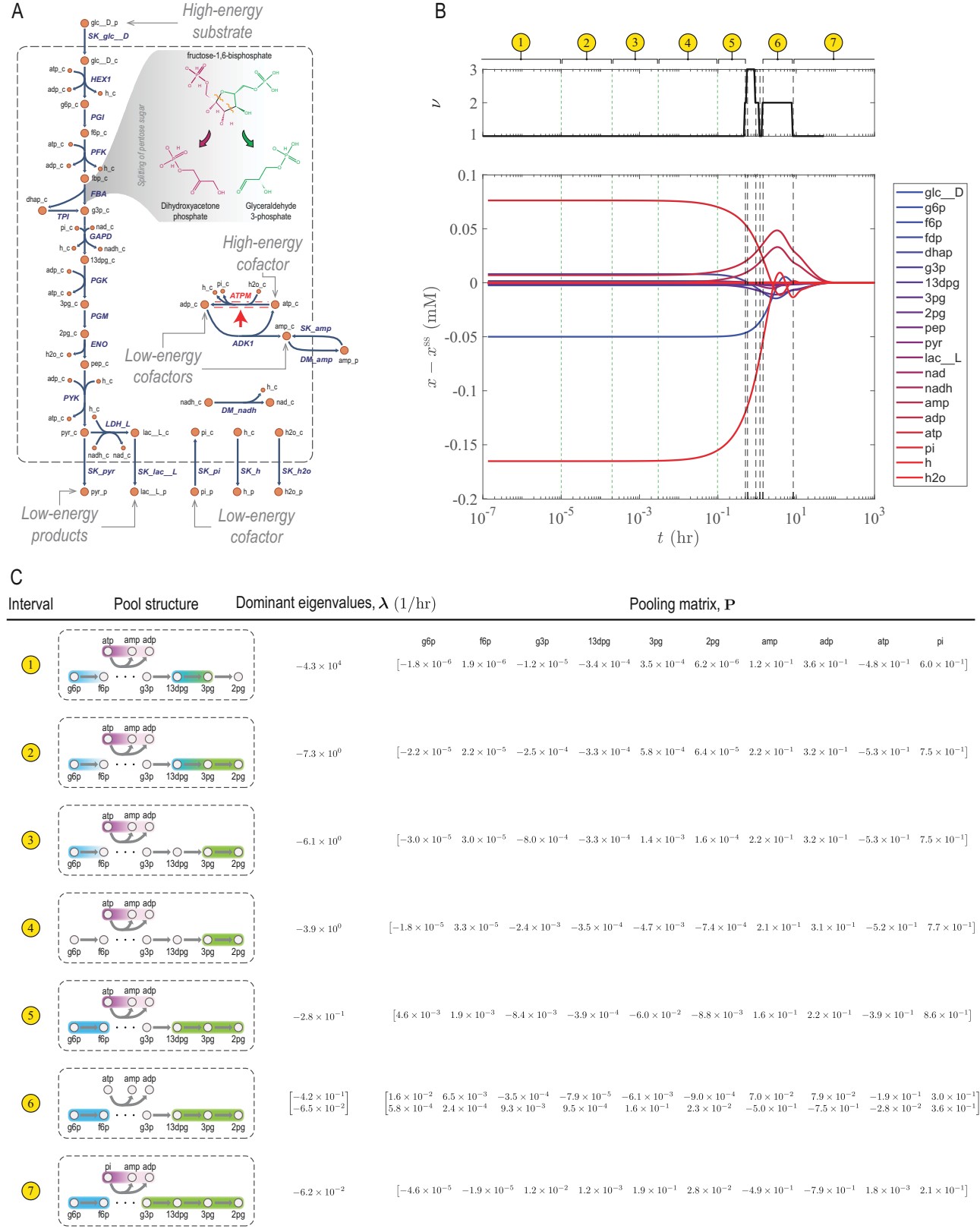

**FIG 4** DMA of the glycolytic pathway. (A) A network map, highlighting the energetics of substrate, products, and cofactors. The flux of the reaction indicated by the red arrow is perturbed. The retro-aldol cleavage of fructose 1,6-bisphosphate is a key step, making the energy stored in the chemical bonds of the

**FIG 4** (Continued)

pentose ring accessible for the production of high-energy cofactors in downstream reactions. (B) Dynamic response to flux perturbations. Deviations from the steady-state $x - x^{ss}$ and dimensionality $\nu$ are plotted along the dynamic trajectory. Dashed and dotted lines indicate a transition between two time periods with distinct exponential decay modes, partitioning the overall relaxation time into seven characteristic time intervals. Dimensionality changes across dashed lines but remains the same across dotted lines. (C) Dominant eigenvalues and pooling matrix for each time interval identified in panel B. Gradient and uniformly colored envelopes in pool structures enclose metabolites, the concentration trajectories of which are negatively and positively correlated in the corresponding time interval, respectively.

The first coherent structure in this case study arises from the relaxation of phosphoglycerate mutase (PGM) on the ~0.1–100 s timescale, where the concentrations of 3-phosphoglycerate (3pg) and 2-phosphoglycerate (2pg) become positively correlated. Next, the flux and concentration disturbances of phosphoglucoisomerase and PGM fully relax on the ~1- to 30-min timescale. Here, the concentrations of glucose 6-phosphate and fructose 6-phosphate become positively correlated, forming the well-known hexose phosphate coherent structure (22). On this timescale, the concentration of 1,3-bisphosphoglycerate also becomes correlated with those of 3pg and 2pg, leading to a larger phosphoglycerate coherent structure. On the ~10-hr timescale, the dynamics of glyceraldehyde phosphate dehydrogenase relax, resulting in glyceraldehyde 3-phosphate merging with the phosphoglycerate coherent structure. All these coherent structures and their respective timescales are consistent with previous studies of the glycolytic dynamics using bottom-up approaches (22).

In general, coherent structures associated with slow timescales are more physiologically relevant than those forming on fast timescales (27). Thus, we studied the two slowest pools of the glycolytic pathway forming in the last two of the 11 time intervals that DMA identified (Fig. 4B, intervals 6 and 7), examining their respective time evolutions $\pi_1(t)$ and $\pi_2(t)$ (Fig. 5). Here, $\pi_1$ and $\pi_2$ are the pools associated the second slowest and slowest timescales, where $\pi_i(t) := \mathbf{p}_i^T \mathbf{x}(t)$ denotes the representation of pool $i$ with respect to concentrations rather than concentration deviations. We found that a balance between high-energy phosphate bond and redox trafficking shapes the interconnected dynamics of upper and lower glycolysis (Fig. 5), highlighting the coupling role of the cofactors.

Finally, we note that the magnitude of the coefficients of the cofactors (AMP, ADP, ATP, and inorganic phosphate) in the pooling matrix is the largest across all timescales, implying that the glycolytic dynamics are mostly determined by the energetics of cofactor interconversions. Importantly, ATP and inorganic phosphate are the high-energy cofactors that control the dynamics on the circulation ($t \lesssim 1$ min) and circadian ($t \gtrsim 10$ hr) timescales, respectively.

## DISCUSSION

Living systems grow and evolve in constantly changing environments, adapting to external fluctuations through interconnected networks of chemical transformations. Understanding how these dynamics emerge from the underlying molecular processes is a major challenge of systems biology. While whole-cell kinetic models offer a thorough description of the intricate interactions within biological networks, their sheer scale often obscures the fundamental patterns that govern the system-level behavior. Using timescale decomposition techniques, we can overcome this limitation by simplifying the complex kinetics into a coarse-grained model with biologically interpretable components. This approach allows us to glean meaningful insights from the model and identify critical components that drive the dynamics at the system level.

In this paper, we introduced DMA—a data-driven approach for timescale decomposition of chemical reaction networks. This approach characterizes concentration pools and coherent structures emerging from network dynamics using time-series data. A key component of our computational framework is an extended version of optimal DMD (ODMD) (18–20), which we developed to identify characteristic timescales

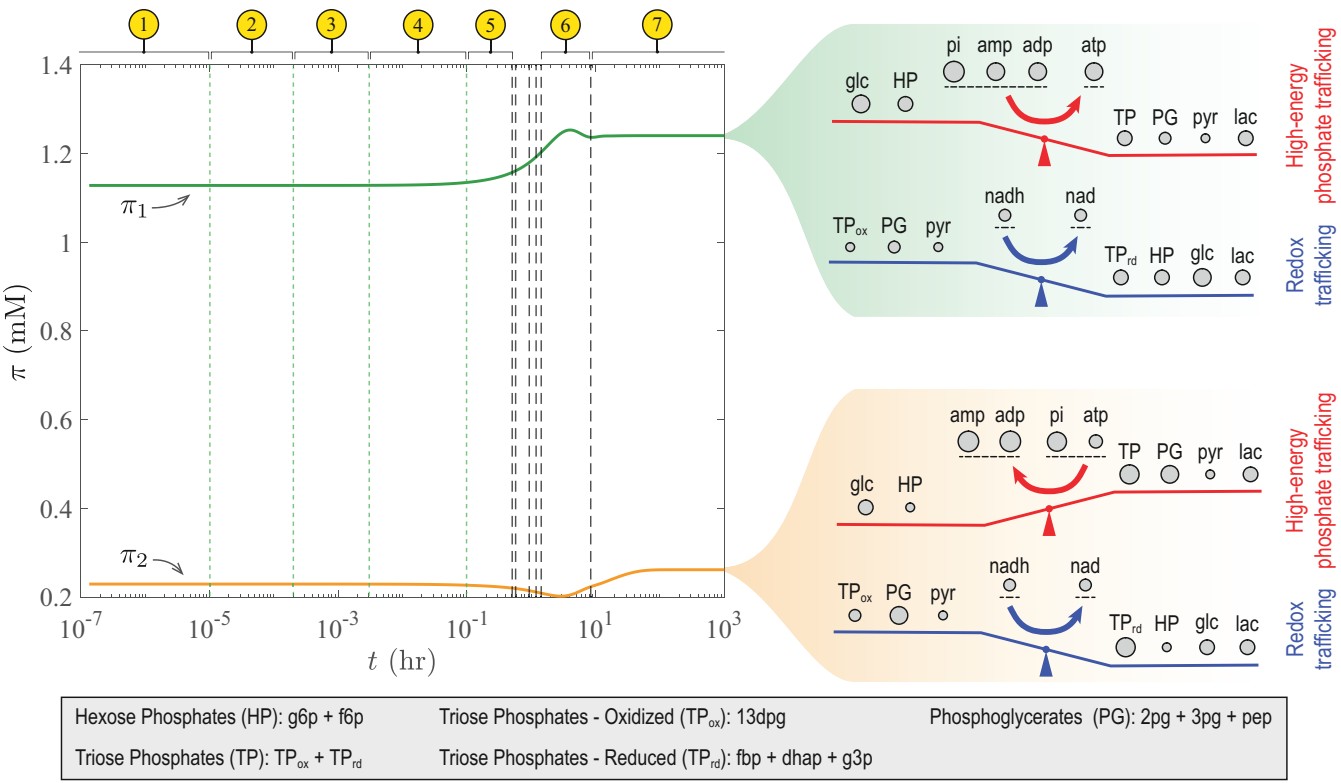

FIG 5  Dynamic trajectories of the slowest pools of glycolysis. The characteristic time intervals are identical to those in Fig. 4B. Note that pools $\pi_1$ and $\pi_2$ are defined with respect to concentrations $\mathbf{x}$ rather than concentration deviations $\chi$. Both pools are dominant in interval 6, and $\pi_2$ is the dominant pool in interval 7. The coherent structures of $\pi_1$ form near the end of interval 6, while those of $\pi_2$ form in interval 7. The dynamics of these pools are driven by a balance between high-energy phosphate bond and redox trafficking. The circle size for each metabolite or aggregate metabolite on the right panel is proportional to its coefficient in the respective pooling map.

of reaction networks. We showed that the dominant eigenvalues and eigenmodes furnished by DMA in any time interval along with dynamic trajectories are determined by two important statistical descriptors, namely, the time-delayed autocorrelation and covariance. Interestingly, the former was a basis of previous top-down approaches for identification of coherent structures in reaction networks (7). Given that timescale hierarchies often underlie the formation of dynamical patterns that lead to dimensionality reduction, our analysis provides a theoretical basis for why time-delayed autocorrelation is an effective metric for statistical analyses of coherent structures.

DMA is a trajectory-based technique that utilizes standard matrix decomposition, eliminating the need to numerically solve nonlinear optimization problems. Thus, it is robust and applicable to large-scale kinetic models. It identifies coherent structures such that they align with a specific exponential decay modes and reflect correlations among concentration trajectories, linking them to the dissipative structure of the reaction system. Therefore, unlike other trajectory-based techniques, how energy is dissipated throughout reaction networks is inferable from the dynamics of coherent structures. Nevertheless, the results can still be affected by noise, especially in transitory regimes where DMA cannot fully decompose the dominant decay modes if timescales are not sufficiently separated. To alleviate the effect of noise, various preprocessing techniques can be applied to the data matrices, such as sparsity-promoting low-rank decomposition, low-rank projection (solving a generalized form of equation 32), and debiased DMD (13).

We validated the results of DMA for a hypothetical pathway with analytically characterized concentration pools. We also reproduced some of the concentration pools

and coherent structures of the glycolytic pathway that were determined previously using bottom-up approaches (22), further confirming the validity of our approach.

Overall, the outcomes of timescale decomposition are more amenable to biological interpretation than the solutions of mass-balance equations. For example, our analysis of the glycolytic pathway in human red blood cells indicated two physiologically relevant characteristics: (i) glycolysis is mostly a cofactor-driven pathway, the dynamics of which are controlled by ATP and inorganic phosphate on the circulation and circadian timescales, respectively, and (ii) the slowest dynamics are dominated by a balance between high-energy phosphate bond and redox trafficking. By chronologically organizing major events along with time evolutions, timescale decomposition could provide physiological insights into the dynamics of more complex biological pathways.

More broadly, analyzing the dynamics of whole-cell networks, such as that of human red blood cells, using the techniques developed here can elucidate important cellular functions associated with coherent structures forming on separate timescales, allowing us to uncover the underlying mechanisms governing various physiological processes. At the fastest timescales, coherent structures reflect rapid chemical reactions that enable essential functions such as oxygen transport and gas exchange, while those forming on intermediate timescales reveal the coordination of major subsystems involved in energy and redox balance, such as glycolysis and the pentose phosphate pathway. At the slowest timescales, coherent structures can explain regulatory processes that maintain homeostasis and facilitate cellular adaptation. Therefore, understanding these hierarchical timescales in human red blood cells not only enhances our fundamental knowledge of cellular biology but also has implications for diseases such as anemia, malaria, and sickle cell disease, where disruptions in these timescales play a critical role.

Quantitative models have been a chief driver of progress in biology in recent years (28–30). Kinetic-based (1, 2, 31, 32) and constraint-based (33–38) approaches have been, and will likely continue to be, instrumental in these developments. Both approaches provide a mechanistic description of cellular functions, but each has its own limitations. Kinetic models are generally complex and require numerous parameters that are subject to large uncertainties, while constraint-based models are not suitable for predicting inherently dynamic phenotypes. The timescale decomposition technique presented in this study can bridge the gap between kinetic- and constraint-based modeling by furnishing a systematic framework for construction of coarse-grained kinetic models (39–41) based on intrinsic timescales of biochemical reaction networks. Such reduced-order models are more tractable than whole-cell models and can capture essential characteristics of biological systems at physiologically relevant timescales.

Personalized medicine is another area where our timescale decomposition technique can play a major role. For example, when individual variations of pathological features or risk for drug side effects manifest in cellular dynamics, studying the timescale hierarchies can help identify the underlying metabolic causes (26). Using kinetic constants as a proxy for individual genotypes in these cases, analyzing the patterns in timescales and coherent structures allows us to understand and classify disease phenotypes through the lens of timescale hierarchies.

## MATERIALS AND METHODS

### Concentration pools and coherent structures

We first introduce the concept of pools and coherent structures for a case study, providing a biological motivation for their definitions. We define these concepts formally in the general case in the next section.

Consider a reaction network involving a single linear pathway with four metabolites and five reactions. We refer to this reaction network as Toy Model 1 (Fig. S1A). The pathway imports metabolite 1 from and exports metabolite 4 into the extracellular environment. Here, $x_1^*$ and $x_4^*$ denote the extracellular concentration of metabolites 1

and 4, respectively. The rate constants of consecutive reactions in this pathway are separated by two orders of magnitude (Fig. S1C), so its dynamics are expected to unfold over separate timescales. Dynamic responses to perturbations of steady states are ascertained from transient mass-balance equations:

$$\frac{d\mathbf{x}}{dt} = \mathbf{Sv}, \qquad \mathbf{x} = \mathbf{x}^0 \quad \text{at} \quad t = 0, \tag{1}$$

with $\mathbf{x}$ the concentration vector, $\mathbf{x}^0$ initial concentration perturbations, $\mathbf{S}$ stoichiometric matrix, flux vector, and $t$ time. Upon perturbations, the dynamics of this system relax over four time intervals associated with three separate timescales $T_{1-3}$ (Fig. S1B, dashed lines). Because of the disparity of rate constants, reactions 1–4 reach their steady states consecutively on their respective timescales. As the dynamics of a given reaction relax, the concentrations of its substrates and products become correlated in the intervals between consecutive timescales. Consequently, trajectories move in a low-dimensional space, suggesting that the dynamics can be adequately described with respect to metabolite pools with correlated concentrations. This is a general behavior that most biochemical reaction networks exhibit, and it is the basis of the definition of concentration pools and coherent structures in this section.

To formalize the concept of concentration pools, it is more convenient to express concentrations and fluxes relative to their steady-state values. Concentration and flux deviations are defined as $\chi := \mathbf{x} - \mathbf{x}^{ss}$ and $\vartheta := \mathbf{v} - \mathbf{v}^{ss}$, respectively. Accordingly, mass-balance equations can be expressed with respect to these deviation variables:

$$\frac{d\chi}{dt} = \mathbf{S}\vartheta, \qquad \chi = \chi^0 \quad \text{at} \quad t = 0. \tag{2}$$

Assuming mass-action kinetics, reaction rates are expressed with respect to deviation variables as follows:

$$\vartheta_1 = -k_1^- \chi_1, \tag{3a}$$
$$\vartheta_2 = k_2^+ \chi_1 - k_2^- \chi_2, \tag{3b}$$
$$\vartheta_3 = k_3^+ \chi_2 - k_3^- \chi_3, \tag{3c}$$
$$\vartheta_4 = k_4^+ \chi_3 - k_4^- \chi_4, \tag{3d}$$
$$\vartheta_5 = k_5^+ \chi_4. \tag{3e}$$

The first pool of Toy Model 1 is associated with the first timescale $T_1 \sim \mathcal{O}(1/k_2^+)$ with $\mathcal{O}$ the order-of-magnitude operator. On this timescale, all flux deviations but $\vartheta_2$ are negligible. Therefore, $\vartheta_2 \gg \vartheta_1, \vartheta_3, \vartheta_4, \vartheta_5$, and the mass-balance equations describing the dynamics are as follows:

$$\frac{d\chi_1}{dt} = -\vartheta_2, \tag{4a}$$
$$\frac{d\chi_2}{dt} = \vartheta_2. \tag{4b}$$

Eliminating $\chi_1$ and $\chi_2$, equations 4a and b can be expressed with respect to $\vartheta_2$ as follows:

$$\frac{d\vartheta_2}{dt} = -(k_2^+ + k_2^-)\vartheta_2 \quad \Rightarrow \quad \vartheta_2(t) = \vartheta_2^0 \exp(\mu_1 t),$$

where $\mu_1 := -k_2^+ - k_2^-$, and $\vartheta_2^0$ is the initial condition for the flux deviation of reaction 2. From this analysis, we derive a more accurate approximation of the first timescale as

$T_1 \sim -1/\mu_1$. Substituting the solution of $\vartheta_2$ back in equations 4a and 4b, we obtain the trajectory of concentration deviations:

$$
\begin{bmatrix} \chi_1 \\ \chi_2 \\ \chi_3 \\ \chi_4 \end{bmatrix} = \vartheta_2^0/\mu_1 \underbrace{\begin{bmatrix} -1 \\ 1 \\ 0 \\ 0 \end{bmatrix}}_{\phi_1} \exp(\mu_1 t) + \begin{bmatrix} c_1 \\ c_2 \\ 0 \\ 0 \end{bmatrix}, \tag{5}
$$

where $\phi_1$ is the eigenmode associated with the first pool in the primal space (see Fig. 1F) with $c_1$ and $c_2$ the integration constants of equations 4a and 4b. We observe from equations 4a and 4b that $\chi_1(t)$ and $\chi_2(t)$ are negatively correlated on this timescale. Once the dynamics of reaction 2 have relaxed, its flux deviation equilibrates so that

$$
\vartheta_2 \to 0 \quad \text{as} \quad t \to T_2 \quad \Rightarrow \quad \chi_2 \to K_2^{\text{eq}}\chi_1 \quad \text{and} \quad \frac{\mathrm{d}\chi_2}{\mathrm{d}t} \to K_2^{\text{eq}}\frac{\mathrm{d}\chi_1}{\mathrm{d}t}, \tag{6}
$$

with $K_2^{\text{eq}} := k_2^+/k_2^-$ the equilibrium constant of reaction 2 and $T_2$ the second timescale. Note that equilibration here refers to the relaxation of flux disturbances as $\vartheta_2 \to 0$, which is not the same as the equilibration of reaction 2 when $v_2 \to 0$.

If the relaxation time of reaction 2 is faster than the second timescale of this system, then there is an intermediate timescale $\check{T}_1 \in [T_1, T_2]$ that characterizes the transition between the first and second timescales during which reaction 2 equilibrates (see Fig. S1B). In this transition period, the flux deviations of reactions 2 and 3 are of the same order, so that $\vartheta_2 \sim \vartheta_3 \gg \vartheta_1, \vartheta_4, \vartheta_5$. To quantify this transitory period, suppose that $\vartheta_2 = \theta\vartheta_3$ with $\theta$ a coefficient that characterizes the transition between the two timescales when $\theta \sim \mathcal{O}(1)$. Accordingly, the mass-balance equations simplify to

$$
\frac{\mathrm{d}\chi_1}{\mathrm{d}t} = -\theta\vartheta_3, \tag{7a}
$$

$$
\frac{\mathrm{d}\chi_2}{\mathrm{d}t} = (\theta - 1)\vartheta_3, \tag{7b}
$$

$$
\frac{\mathrm{d}\chi_3}{\mathrm{d}t} = \vartheta_3. \tag{7c}
$$

From the asymptotic relation between $\mathrm{d}\chi_2/\mathrm{d}t$ and $\mathrm{d}\chi_1/\mathrm{d}t$ stated in equation 6, we find that equations 7a–7c with $\theta = \theta^{\text{a}} := 1/(1 + K_2^{\text{eq}})$ approximate the dynamics in this transitory regime when the second timescale of the system is approached. Note that, in the transitory interval, the coefficient $\theta$ in these equations varies from a large value $\theta \gg 1$ near the first timescale to its asymptotic value $\theta^{\text{a}}$ near the second timescale. However, we treat it as a constant to approximate local solutions of equations 7a–7c in the transitory regime. We can eliminate $\chi_2$ and $\chi_3$ in equations 7b and 7c in the same way as in equations 4a and 4b to express the mass-balance equations with respect to $\vartheta_3$:

$$
\frac{\mathrm{d}\vartheta_3}{\mathrm{d}t} = \left[(\theta - 1)k_3^+ - k_3^-\right]\vartheta_3 \quad \Rightarrow \quad \vartheta_3(t) = \vartheta_3^0 \exp(\check{\mu}_1 t),
$$

where $\check{\mu}_1 := (\theta - 1)k_3^+ - k_3^-$ and $\vartheta_3^0$ is the initial condition for the flux deviation of reaction 3. The timescale associated with this regime is estimated as $\check{T}_1 \sim -1/\check{\mu}_1$, and the trajectory of concentration deviations is ascertained by integrating equations 7a–7c using the solution of $\vartheta_3$:

$$\begin{bmatrix} \chi_1 \\ \chi_2 \\ \chi_3 \\ \chi_4 \end{bmatrix} = \vartheta_3^0/\check{\mu}_1 \underbrace{\begin{bmatrix} -\theta \\ \theta - 1 \\ 1 \\ 0 \end{bmatrix}}_{\check{\phi}_1} \exp(\check{\mu}_1 t) + \begin{bmatrix} c_1 \\ c_2 \\ c_3 \\ 0 \end{bmatrix},$$ (8)

where $c_{1-3}$ are integration constants. Note that $K_2^{eq} = 1$ (Fig. S1C) and $\theta^a = 1/2$ in Toy Model 1, implying that $\chi_1(t)$ and $\chi_2(t)$ are positively correlated, and, together, they are negatively correlated with $\chi_3(t)$ on this timescale. Finally, we highlight an important feature of dominant decay modes during the transition between two consecutive timescales in relation to the role of kinetic parameters. Consider the timescales $T_1$ and $\check{T}_1$ of Toy Model 1, for example. Here, the coefficients of $\chi_1$ and $\chi_2$ in the decay mode $\phi_1$ only depend on their stoichiometric coefficients in reaction 2. However, in the decay mode $\check{\phi}_1$, they depend on both the stoichiometric coefficients and rate constants $k_2^+$ and $k_2^-$.

Having qualitatively described how dynamic trajectories become correlated on the first timescale, we provide a quantitative representation of the first pool. In this work, we define a pool associated with a given timescale as a linear combination of concentrations, the time-dependent representation of which aligns with the corresponding exponential decay. If multiple dominant timescales coexist in a time interval, we require the corresponding pools to be dynamically independent. For example, in the transitory period outlined above where both exponential decay modes in equations 5 and 8 are dominant, the first pool $p_1$ and its transitory counterpart $\check{p}_1$ are defined as follows:

$$p_1(t) := \mathbf{p}_1^T \chi(t) = (K_2^{eq} + 1)\vartheta_2^0/\mu_1 \exp(\mu_1 t) + \mathcal{R}_1(t),$$ (9a)

$$\check{p}_1(t) := \check{\mathbf{p}}_1^T \chi(t) = 3\vartheta_3^0/\check{\mu}_1 \exp(\check{\mu}_1 t) + \check{\mathcal{R}}_1(t),$$ (9b)

where

$$\mathbf{p}_1 := \left[-K_2^{eq}, 1, 0, 0\right]^T,$$ (10a)

$$\check{\mathbf{p}}_1 := \left[-1, -1, 2, 0\right]^T$$ (10b)

are normalized vector representations of the pools. On occasion, we also refer to these vector representations as pooling maps. Here, $\mathcal{R}_1(t)$ and $\check{\mathcal{R}}_1(t)$ are the residual terms approaching a steady-state value at a faster rate than the exponential decay term for each pool. Moreover, the timescale associated with each pool is defined as a time point beyond which the pool is sufficiently close to its steady-state value (Fig. S2A). For example, the timescales of the foregoing two pools are defined as follows:

$$T_1 := T \quad \text{such that} \quad |p_1(t) - p_1(t \to \infty)| < \varepsilon_p \mathcal{C}_1 \quad \forall t \geq T,$$ (11a)

$$\check{T}_1 := T \quad \text{such that} \quad |\check{p}_1(t) - \check{p}_1(t \to \infty)| < \varepsilon_p \check{\mathcal{C}}_1 \quad \forall t \geq T.$$ (11b)

Here, $\varepsilon_p$ is a tolerance threshold controlling the closeness of pools to their steady-state values with $\mathcal{C}_1$ and $\check{\mathcal{C}}_1$ appropriate concentration scales. For example, for $p_1$ and $\check{p}_1$, $\mathcal{C}_1 = \vartheta_2^0/\mu_1$ and $\check{\mathcal{C}}_1 = \vartheta_3^0/\check{\mu}_1$ are reasonable concentration scales.

As noted in previous works, a proper definition of pools should ensure that each pool is dynamically independent of other pools in a system when they form on the same timescale (22, 27). In top down approaches, such as principal component analysis or independent component analysis, orthogonality and independence are equivalent concepts (42). However, as discussed at the beginning of this section, a biological motivation for introducing the concept of concentration pools is identifying aggregate variables that naturally arise in systems with low-dimensional dynamics—a key

characteristic of reaction networks with timescale hierarchies. Accordingly, alignment with exponential decay modes associated with separate timescales is considered a more appropriate notion of independence (18, 22). Since eigenmodes are not necessarily orthogonal for a general reaction network, dynamical independence does not imply the orthogonality of pools in the concentration space. In this work, we adopt the same notion of independence. However, in addition to alignment with exponential decay modes, we also require a reciprocal orthogonality between the vector representation of the pools and eigenmodes. Here, we present this reciprocal relationship for $p_1$ and $\check{p}_1$ in the transitory regime between the first two timescales of Toy Model 1, deferring the formulation for the general case to the next section. The reciprocal orthogonality conditions for the first pool and its transitory counterpart are as follows:

$$\mathbf{p}_1^{\mathrm{T}}\check{\boldsymbol{\phi}}_1 = 0, \tag{12a}$$
$$\check{\mathbf{p}}_1^{\mathrm{T}}\boldsymbol{\phi}_1 = 0. \tag{12b}$$

These conditions are motivated by their important geometric and physical interpretations, and they are linked to the concept of flux-concentration duality (43) in chemical reaction networks (see "Reciprocal orthogonality conditions for Toy Model 1" and Fig. 1F).

The transitory regime between two consecutive timescales generally have two stages. In the first, the concentrations evolve in a direction corresponding to a maximal energy dissipation of the reaction, where the substrate and product concentrations are negatively correlated. In the second, the concentrations evolve in a direction associated with the equilibration of the flux disturbance, where the substrate and product concentrations are positively correlated. We refer to the first as the "disequilibrium" and the second as the "conservation" stages of relaxation for each reaction.

Coherent structure is another concept that we examine in this paper. Its definition, which is derived from, but is more restrictive than that of concentration pools, centers around correlations among concentration trajectories. As we noted for Toy Model 1, the dominant eigenmodes on a given timescale span a reduced concentration space in which the dynamics unfold. Accordingly, metabolite coefficients in the pooling maps, which are in turn ascertained from the respective eigenmodes, determine metabolites the concentrations of which are affected on that timescale. The concentration trajectories of these metabolites remain correlated until the next timescale of the system has been reached. For a linear system, such as Toy Model 1, the eigenmodes and their respective pooling maps do not vary with time. Therefore, the correlation coefficients among metabolites that are present in a pool remain constant. However, in a general nonlinear system, the Jacobian matrix and its eigenmodes can vary along dynamic trajectories. Consequently, correlation coefficients also become time-dependent, but they can plateau in small intervals between the timescales of the system.

For a general nonlinear reaction system, we define a coherent structure as a subset of metabolites in a concentration pool, the correlation coefficients of which vary within a prescribed tolerance threshold in time intervals between the timescales of the system. To make this definition more precise, suppose that metabolites $i$ and $j$ in a reaction system are part of a pool that becomes active in the interval $[T_1, T_2]$, where $T_1$ and $T_2$ are two consecutive timescales. Then, for these metabolites to form a coherent structure, their concentration trajectories must satisfy

$$|\rho(x_i, x_j) - 1| < \varepsilon_\rho, \tag{13}$$

where $\varepsilon_\rho$ is a tolerance threshold controlling variations of correlation coefficients, and the correlation function $\rho$ is given by

$$\rho(x_i, x_j) := \frac{\mathbb{E}(x_i x_j) - \mathbb{E}(x_i)\mathbb{E}(x_j)}{\sigma(x_i)\sigma(x_j)}, \tag{14a}$$

$$\mathbb{E}(f) := \frac{1}{t^2 - t^1} \int_{t^1}^{t^2} f(t)\mathrm{d}t, \tag{14b}$$

$$\sigma(f) := \sqrt{\mathbb{E}(f^2) - \mathbb{E}(f)^2}. \tag{14c}$$

Here, $\mathbb{E}(f)$ and $\sigma(f)$ are the expectation and standard deviation of a time-dependent function $f(t)$. Moreover, the correlation function is evaluated in the interval $\left[t^1, t^2\right]$ in which the coherent structure forms, where $T_1 \leq t^1 < t^2 \leq T_2$. Note that superscripts of $t$ in equations 14a–14c are indices referring to time-interval bounds and should not be confused with exponents. For coherent structures with more than two metabolites, equation 13 must be satisfied for all pairwise correlations among the metabolites. For linear systems, such as Toy Model 1, concentration pools always form a coherent structure because the pooling maps and correlation coefficients are constant along dynamic trajectories. Also note that coherent structures can generally span several timescales.

Having discussed the concepts of concentration pools and coherent structures for the first timescale of Toy Model 1, we follow the same procedure to characterize concentration pools and coherent structures that form at slower timescales. The second pool arises when the dynamics of reaction 3 begin to relax. It is associated with the timescale $T_2 \sim -1/\mu_2$ with $\mu_2 := (\theta^a - 1)k_3^+ - k_3^-$. On this timescale, we have $\vartheta_2 \simeq \theta^a \vartheta_3 \gg \vartheta_1, \vartheta_4, \vartheta_5$, leading to the following expression for the second pool:

$$p_2'(t) := \mathbf{p}_2'^{\mathrm{T}}\chi(t) = \left(\frac{K_2^{\mathrm{eq}}K_3^{\mathrm{eq}} + K_2^{\mathrm{eq}} + 1}{1 + K_2^{\mathrm{eq}}}\right)\frac{\vartheta_3^0}{\mu_2}\exp(\mu_2 t) + \mathcal{R}_2'(t), \tag{15}$$

$$\mathbf{p}_2' := \left[0, -K_3^{\mathrm{eq}}, 1, 0\right]^{\mathrm{T}}. \tag{16}$$

Here, $\mathcal{R}_2'(t)$ denotes a residual term that approaches a steady-state value at a faster rate than the exponential decay term, so its leading-order term is $\exp(\mu_1 t)$. Another expression for the second pool can be derived by combining $p_2'$ with pools that form on faster timescales than $-1/\mu_2$. For example, once the dynamics of reaction 2 have relaxed in the transitory interval $\left[\check{T}_1, T_2\right]$, $\chi_1(t)$ and $\chi_2(t)$ become correlated through the relationship:

$$\chi_2(t) = K_2^{\mathrm{eq}}\chi_1(t) + r_1(t), \tag{17}$$

where $r_1(t)$ is a residual, the leading-order term of which is $\exp(\mu_1 t)$. Substituting equation 17 in equation 15 results in the following:

$$p_2''(t) := \mathbf{p}_2''^{\mathrm{T}}\chi(t) = \left(\frac{K_2^{\mathrm{eq}}K_3^{\mathrm{eq}} + K_2^{\mathrm{eq}} + 1}{1 + K_2^{\mathrm{eq}}}\right)\frac{\vartheta_3^0}{\mu_2}\exp(\mu_2 t) + \mathcal{R}_2''(t), \tag{18}$$

$$\mathbf{p}_2'' := \left[-K_2^{\mathrm{eq}}K_3^{\mathrm{eq}}, 0, 1, 0\right]^{\mathrm{T}}. \tag{19}$$

A sum-total pool can also be constructed by adding the first two expressions in equations 15 and 18, resulting in

$$p_2(t) := \mathbf{p}_2^{\mathrm{T}}\chi(t) = p_2'(t) + p_2''(t) = 2\left(\frac{K_2^{\mathrm{eq}}K_3^{\mathrm{eq}} + K_2^{\mathrm{eq}} + 1}{1 + K_2^{\mathrm{eq}}}\right)\frac{\vartheta_3^0}{\mu_2}\exp(\mu_2 t) + \mathcal{R}_2(t), \tag{20}$$

$$\mathbf{p}_2 := \mathbf{p}_2' + \mathbf{p}_2'' = \left[-K_2^{\mathrm{eq}}K_3^{\mathrm{eq}}, -K_3^{\mathrm{eq}}, 2, 0\right]^{\mathrm{T}}, \tag{21}$$

with $\mathcal{R}_2(t)$ the sum-total residual term. Note that the leading-order terms of $\mathcal{R}'_2(t)$, $\mathcal{R}''_2(t)$, and $\mathcal{R}_2(t)$ are $\exp(\mu_1 t)$. Thus, $p'_2$, $p''_2$, and $p_2$ can be regarded as three representations of the second pool since they all align with $\exp(\mu_2 t)$, exhibiting a similar asymptotic behavior for $t \gtrsim T_2$ (Fig. S2B). Note that the pooling maps for all three representations are orthogonal to the second transitory mode that becomes active in the interval $\left[\check{T}_2, T_3\right]$, so they satisfy the reciprocal orthogonality condition:

$$\mathbf{p}_2'^{\mathrm{T}}\check{\boldsymbol{\phi}}_2 = \mathbf{p}_2''^{\mathrm{T}}\check{\boldsymbol{\phi}}_2 = \mathbf{p}_2^{\mathrm{T}}\check{\boldsymbol{\phi}}_2 = 0, \tag{22}$$

which is similar to the reciprocal orthogonality condition in equation 12a for the first timescale.

The third timescale $T_3$ is associated with the relaxation of reaction 4. The procedure for characterizing its concentration pool is similar to that outlined for the second timescale (see "Derivation of the third and fourth approximate eigenmodes of Toy Model 1"). We only highlight the representations of the third pooling map:

$$\mathbf{p}'_3 := \left[0, 0, -K_4^{\mathrm{eq}}, 1\right]^{\mathrm{T}}, \tag{23a}$$

$$\mathbf{p}''_3 := \left[0, -K_3^{\mathrm{eq}}K_4^{\mathrm{eq}}, 0, 1\right]^{\mathrm{T}}, \tag{23b}$$

$$\mathbf{p}'''_3 := \left[-K_2^{\mathrm{eq}}K_3^{\mathrm{eq}}K_4^{\mathrm{eq}}, 0, 0, 1\right]^{\mathrm{T}}, \tag{23c}$$

$$\mathbf{p}_3 := \mathbf{p}'_3 + \mathbf{p}''_3 + \mathbf{p}'''_3 = \left[-K_2^{\mathrm{eq}}K_3^{\mathrm{eq}}K_4^{\mathrm{eq}}, -K_3^{\mathrm{eq}}K_4^{\mathrm{eq}}, -K_4^{\mathrm{eq}}, 3\right]^{\mathrm{T}}. \tag{23d}$$

The corresponding pools $p'_3$, $p''_3$, $p'''_3$, and $p_3$ exhibit a similar asymptotic behavior for $t \gtrsim T_3$ (Fig. S2C). As with the second timescale, $\mathbf{p}_3$ denotes the sum-total representation of the third pooling map.

The dynamics of reaction 4 relax fully once the initial perturbations have propagated throughout the network and reached the boundary reactions. Upon relaxation, all flux and concentration disturbances equilibrate toward a steady state on a transitory timescale $\check{T}_3$. On this timescale, which we regard as the fourth timescale of Toy Model 1 (i.e., $T_4 := \check{T}_3$), all the concentration deviations align with the slowest eigenmode, forming a concentration pool containing all the intracellular metabolites. Since this pool corresponds to the only dominant eigenmode for $t \gtrsim T_4$, there are no slower eigenmodes to evolve into; hence, it needs not satisfy any reciprocal orthogonality conditions. Accordingly, the concentration pool associated with the largest timescale is defined as a normalized form of the slowest eigenmode. This condition automatically arises from the general definition of the pooling matrix—a matrix containing all the dominant pooling maps on a given timescale—which will be formalized for a general nonlinear reaction system in the next section.

The fourth pool has the following representations (see "Derivation of the third and fourth approximate eigenmodes of Toy Model 1"):

$$\mathbf{p}'_4 := [1, 0, 0, 0]^{\mathrm{T}}, \tag{24a}$$

$$\mathbf{p}''_4 := [0, K_2^{\mathrm{eq}}, 0, 0]^{\mathrm{T}}, \tag{24b}$$

$$\mathbf{p}'''_4 := [0, 0, K_2^{\mathrm{eq}}K_3^{\mathrm{eq}}, 0]^{\mathrm{T}}, \tag{24c}$$

$$\mathbf{p}''''_4 := [0, 0, 0, K_2^{\mathrm{eq}}K_3^{\mathrm{eq}}K_4^{\mathrm{eq}}]^{\mathrm{T}}, \tag{24d}$$

$$\mathbf{p}_4 := \mathbf{p}'_4 + \mathbf{p}''_4 + \mathbf{p}'''_4 + \mathbf{p}''''_4 = \left[1, K_2^{\mathrm{eq}}, K_2^{\mathrm{eq}}K_3^{\mathrm{eq}}, K_2^{\mathrm{eq}}K_3^{\mathrm{eq}}K_4^{\mathrm{eq}}\right]^{\mathrm{T}}. \tag{24e}$$

The asymptotic profiles of the respective pools $p'_4$, $p''_4$, $p'''_4$, $p''''_4$, and $p_4$ are similar for $t \gtrsim T_4$ (Fig. S2D). As before, $\mathbf{p}_4$ denotes a sum-total representation.

Lastly, we emphasize that the purpose of the analysis presented here and subsequent sections (see "Derivation of the third and fourth approximate eigenmodes of Toy Model 1") is to elucidate the connections between concentration pools, timescales, and the

equilibration of flux disturbances using approximate methods. This approach is justified since a fundamental characteristic of biochemical reaction networks underlying the separation of timescales is the presence of vastly disparate rate constants. The solutions provided here and in subsequent sections (see "Derivation of the third and fourth approximate eigenmodes of Toy Model 1") are not exact, so the eigenvalues $\mu_i$ estimated here should be regarded as approximations of the exact eigenvalues $\lambda_i$. Since Toy Model 1 is a linear system, its eigenvalues remain constant along dynamic trajectories. Consequently, in any time interval, the exact solution of mass-balance equations can be expressed as a linear combination of exponential decay modes with varying amplitudes without needing transitory eigenvalues. However, the reason why we encountered the transitory eigenvalues $\breve{\mu}_i$ here lies in our approximation method describing the kinetics on a given timescale by the flux of a single reaction, the rate constant of which gives rise to that timescale.

## Dynamic mode analysis

DMA is a data-driven approach to identify timescale hierarchies of a chemical reaction network and characterize its dynamic responses to flux or concentration perturbations (Fig. 1). Specifically, this approach aims to compute concentration pools and coherent structures that emerge as a reaction network evolves in time from time-series data. Although data can be generated from experimental measurements or numerical simulations, we only focus on numerical solutions of kinetic models of biochemical reaction networks in this paper. We assume that the concentration trajectories of all the metabolites in the network of interest are given. The goal is then to identify the dominant eigenvalues, dominant eigenmodes, and the respective concentration pools algorithmically in any time interval along dynamic trajectories of the network.

The algorithm begins by solving the mass-balance equations (equation 1) for a reaction network with $n$ metabolites and $m$ reactions, the steady-state concentrations or fluxes of which is perturbed (Fig. 1A). The resulting concentration trajectory $\mathbf{x}(t)$ is then computed numerically (Fig. 1B). The goal in subsequent steps of DMA is to identify the dominant eigenvalues and eigenmodes in a sliding time window spanning the interval $[t^1, t^2]$ as it moves from $t = 0$ to $t = T_\infty$, where $t = T_\infty$ is the total relaxation time—a time by which all flux and concentration disturbances have relaxed to within a tolerable threshold. To analyze the dynamics in the sliding time window, the mass-balance equations are linearized locally around a reference time $t^\diamond \in [t^1, t^2]$:

$$\frac{d\mathbf{h}}{dt} = \mathbf{Jh} + \boldsymbol{a}, \qquad \mathbf{h} = \mathbf{h}^0 \quad \text{at} \quad t = 0, \tag{25}$$

where

$$\mathbf{f} := \mathbf{Sv}, \quad \mathbf{J} := \left(\frac{\partial \mathbf{f}}{\partial \mathbf{x}}\right)_{\mathbf{x}^\diamond} = \mathbf{SG}, \quad \mathbf{G} := \left(\frac{\partial \mathbf{v}}{\partial \mathbf{x}}\right)_{\mathbf{x}^\diamond}, \quad \boldsymbol{a} := \mathbf{f}(\mathbf{x}^\diamond).$$

Here, $\mathbf{J}$ is the local Jacobian matrix, $\mathbf{x}^\diamond := \mathbf{x}(t^\diamond)$ the concentration vector at the reference time, and $\mathbf{h} := \mathbf{x} - \mathbf{x}^\diamond$ the local concentration deviations. Note that, for a general nonlinear reaction network, equation 25 is an inhomogeneous system of equations, the solutions of which cannot be expressed with respect to purely exponential decay terms (Fig. 1C). Thus, we recast it into the homogenous system:

$$\frac{d\widetilde{\mathbf{h}}}{dt} = \mathbf{J}\widetilde{\mathbf{h}}, \qquad \widetilde{\mathbf{h}} = \widetilde{\mathbf{h}}^0 \quad \text{at} \quad t = 0, \tag{26}$$

by introducing a new variable $\widetilde{\mathbf{h}} := \mathbf{h} + \mathbf{w}$, we term the local transformed concentration deviation, where $\mathbf{w} := \mathbf{J}^{-1}\boldsymbol{a}$ captures the inhomogeneity of equation 25. The general solution of equation 26 is written (44) as follows:

$$\widetilde{\mathbf{h}}(t) = \mathrm{Exp}(\mathbf{J}t)\widetilde{\mathbf{h}}^0, \tag{27}$$

where Exp is the exponential map. Since we are only concerned with small perturbations of noncritical stable steady states, we assume that $\mathfrak{Re}(\lambda) \S lt; \mathbf{0}$ along all dynamic trajectories. Therefore, $\mathbf{J}$ is always nonsingular, so $\mathbf{w}$ is a well-defined quantity. Here, $\mathfrak{Re}(\,\cdot\,)$ returns the real part of a complex argument, and $\lambda$ is the eigenvalue vector of the Jacobian. The linearization of mass-balance equations introduced here allows the application of top-down timescale decomposition techniques (e.g., DMD) to identify timescale hierarchies from time-series data.

In the next step, time-series data are generated from the numerical solution of local concentration deviations in the current time window by evaluating $\mathbf{h}(t)$ at $N + 1$ equally spaced time points in $[t^1, t^2]$ (Fig. 1D), compiling their values in two data matrices:

$$\mathbf{H}_0 := \begin{bmatrix} | & | & & | \\ \mathbf{h}_1 \mathbf{h}_2 & \cdots & \mathbf{h}_N \\ | & | & & | \end{bmatrix} \in \mathbb{R}^{n \times N}, \tag{28a}$$

$$\mathbf{H}_1 := \begin{bmatrix} | & | & & | \\ \mathbf{h}_2 \mathbf{h}_3 & \cdots & \mathbf{h}_{N+1} \\ | & | & & | \end{bmatrix} \in \mathbb{R}^{n \times N}, \tag{28b}$$

where $\mathbf{h}_k := \mathbf{h}(t_k)$ is the vector of local concentration deviations evaluated at the $k$th time point in $[t^1, t^2]$ with $\Delta t := (t^2 - t^1)/(N - 1)$ the gap between consecutive time points. The corresponding transformed data matrices $\widetilde{\mathbf{H}}_0$ and $\widetilde{\mathbf{H}}_1$ are similarly defined in terms of $\widetilde{\mathbf{h}}_k$. From equation 27, we find the relationship between transformed concentration deviations at two consecutive time points:

$$\widetilde{\mathbf{h}}_{k+1} = \mathbf{A}\widetilde{\mathbf{h}}_k + \epsilon_k, \quad k = 1 \cdots N, \quad \mathbf{A} := \mathrm{Exp}(\mathbf{J}\Delta t). \tag{29}$$

Here, $\epsilon_k$ is an error vector associated with the linearization of mass-balance equations. It also accounts for the errors arising from numerical simulation (e.g., truncation errors) or experimental measurements, depending on how data are generated. Since $\mathbf{A}$ is a constant matrix, we can write equation 29 in a matrix form:

$$\widetilde{\mathbf{H}}_1 = \mathbf{A}\widetilde{\mathbf{H}}_0 + \boldsymbol{\mathcal{E}}, \tag{30}$$

where $\boldsymbol{\mathcal{E}}$ is a matrix of the same size as $\widetilde{\mathbf{H}}_0$ and $\widetilde{\mathbf{H}}_1$, containing all the error vectors $\epsilon_k$.

If mass-balance equations are known, or there are approximation methods to estimate $\mathbf{w}$ from time-series data, then the dominant eigenvalues and eigenmodes of the reaction network in the current time window can be directly determined from $\widetilde{\mathbf{H}}_0$ and $\widetilde{\mathbf{H}}_1$ using top-down approaches, such as DMD (18) or ODMD (19, 20). In the following, we briefly outline the procedure for ODMD.

To determine the dynamic dimensionality of the system, the singular-value decomposition of $\widetilde{\mathbf{H}}_0$ is computed:

$$\widetilde{\mathbf{H}}_0 = \widetilde{\mathbf{U}}\widetilde{\boldsymbol{\Sigma}}\widetilde{\mathbf{V}}^{\mathrm{T}}, \quad \widetilde{\mathbf{U}} \in \mathbb{R}^{n \times \nu}, \quad \widetilde{\boldsymbol{\Sigma}} \in \mathbb{R}^{\nu \times \nu}, \quad \widetilde{\mathbf{V}} \in \mathbb{R}^{N \times \nu},$$

where $\nu$ is the number of singular values that are nonzero to within a tolerable threshold $\varepsilon_{\mathrm{SVD}}$. The columns of $\widetilde{\mathbf{U}}$ are the proper orthogonal modes (20) of $\widetilde{\mathbf{H}}_0$, and they

span a reduced $\nu$-dimensional subspace of the locally transformed concentration-deviation space where the dynamic trajectories lie in the current time window. We refer to this subspace as the space of transformed proper orthogonal modes. The projection of $\mathbf{A}$ onto this reduced space is

$$\mathbf{F} = \widetilde{\mathbf{U}}^{\mathrm{T}}\mathbf{A}\widetilde{\mathbf{U}}. \tag{31}$$

The projected matrix $\mathbf{F} \in \mathbb{R}^{\nu \times \nu}$ in this equation can be proven to be an upper Hessenberg matrix (45). Because $\mathbf{F}$ and $\mathbf{A}$ are related through a similarity transformation, $\mathbf{F}$ contains the dominant eigenvalues of $\mathbf{A}$ in the current time window. Errors in equation 30 are then minimized by solving the optimization problem:

$$\widehat{\mathbf{F}} := \arg \min_{\mathbf{F}} \|\boldsymbol{\mathcal{E}}\|_F^2 = \arg \min_{\mathbf{F}} \|\widetilde{\mathbf{H}}_1 - \mathbf{A}\widetilde{\mathbf{H}}_0\|_F^2 \tag{32}$$

to identify a matrix $\widehat{\mathbf{F}}$ that best describes the data in $\widetilde{\mathbf{H}}_0$ and $\widetilde{\mathbf{H}}_1$, where $\|\cdot\|_F$ denotes the Frobenius norm. Substituting $\mathbf{A}$ from equation 31 in equation 32, the solution of the foregoing optimization problem is ascertained (20) as follows:

$$\widehat{\mathbf{F}} = \widetilde{\mathbf{U}}^{\mathrm{T}}\widetilde{\mathbf{H}}_1\widetilde{\mathbf{V}}\widetilde{\boldsymbol{\Sigma}}^{\dagger}, \tag{33}$$

where the superscript $\dagger$ denotes the Moore-Penrose generalized inverse (21). The eigenmodes of $\widehat{\mathbf{F}}$ are the projections of the dominant eigenmodes of $\mathbf{A}$ in the space of transformed proper orthogonal modes, and its eigenvalues are the dominant eigenvalues of $\mathbf{A}$.

In this paper, both $\mathbf{F}$ and $\mathbf{w}$ are treated as unknowns, so $\widetilde{\mathbf{H}}_0$ and $\widetilde{\mathbf{H}}_1$ are not provided at the outset. Therefore, in the following, we introduce an extension of ODMD (see Fig. S4) to determine the dominant eigenvalues and eigenmodes of the inhomogeneous system (equation 25) directly from the time-series data in $\mathbf{H}_0$ and $\mathbf{H}_1$. We start by rewriting equations 29 and 30 in terms of local concentration deviations:

$$\mathbf{h}_{k+1} = \mathbf{A}\mathbf{h}_k + \boldsymbol{\omega} + \boldsymbol{\epsilon}_k, \quad k = 1 \cdots N, \quad \boldsymbol{\omega} := \mathbf{A}\mathbf{w} - \mathbf{w}, \tag{34}$$

$$\mathbf{H}_1 = \mathbf{A}\mathbf{H}_0 + \mathbf{W} + \boldsymbol{\mathcal{E}}, \quad \mathbf{W} := \boldsymbol{\omega}\mathbf{1}^{\mathrm{T}}, \tag{35}$$

where $\mathbf{1}$ is an all-one column vector with $N$ components. As previously stated, because of the inhomogeneous term $\boldsymbol{\omega}$ in equation 34 or $\mathbf{W}$ in equation 35, DMD or ODMD is not directly applicable, although we can show that $\boldsymbol{\omega} \to \mathbf{0}$ in the limit $\Delta t \to 0$ (see "Inhomogeneity of evolution equations for local concentration deviations and temporal grid size"). Thus, the inhomogeneity can theoretically be eliminated from equation 34 by generating an arbitrarily fine temporal grid in the sliding time window. However, it is impractical to do so because of the computational costs and additional errors it introduces in subsequent steps of the algorithm. Following the same procedure as DMD, the singular-value decomposition of $\mathbf{H}_0$ is computed:

$$\mathbf{H}_0 = \mathbf{U}\boldsymbol{\Sigma}\mathbf{V}^{\mathrm{T}}, \quad \mathbf{U} \in \mathbb{R}^{n \times \nu}, \quad \boldsymbol{\Sigma} \in \mathbb{R}^{\nu \times \nu}, \quad \mathbf{V} \in \mathbb{R}^{N \times \nu}.$$

As in the previous case, $\nu$ reflects the dynamic dimensionality of the system in the current time window, and the columns of $\mathbf{U}$ are the proper orthogonal modes of $\mathbf{H}_0$, spanning the local concentration-deviation space. We refer to this subspace as the space of proper orthogonal modes. As before, $\mathbf{A}$ is expressed with respect to the proper orthogonal modes through the similarity transformation:

$$\mathbf{F} = \mathbf{U}^{\mathrm{T}}\mathbf{A}\mathbf{U}. \tag{36}$$

The goal now is to find $\mathbf{F}$ and $\omega$ such that the errors in equation 35 are minimized. The corresponding optimization problem is written as follows:

$$(\widehat{\mathbf{F}}, \widehat{\omega}) := \arg \min_{(\mathbf{F}, \omega)} \|\boldsymbol{\mathcal{E}}\|_F^2 = \arg \min_{(\mathbf{F}, \omega)} \|\mathbf{H}_1 - \mathbf{A}\mathbf{H}_0 - \mathbf{W}\|_F^2. \tag{37}$$

This is an unconstrained quadratic program, so its solution can be determined analytically (see "Relationship between time-delayed autocorrelation, covariance, and local Jacobian spectra" for the proof):

$$\widehat{\mathbf{F}} = \bar{\mathbf{X}}_{10} \bar{\mathbf{X}}_0^{-1}, \tag{38a}$$

$$\widehat{\omega} = \mathbf{h}_1^{\mathrm{av}} - \mathbf{A}\mathbf{h}_0^{\mathrm{av}}, \tag{38b}$$

where

$$\bar{\mathbf{X}}_{10} := \frac{1}{N} \bar{\mathbf{H}}_1 \bar{\mathbf{H}}_0^{\mathrm{T}} - \bar{\mathbf{h}}_1^{\mathrm{av}} (\bar{\mathbf{h}}_0^{\mathrm{av}})^{\mathrm{T}}, \tag{39a}$$

$$\bar{\mathbf{X}}_0 := \frac{1}{N} \bar{\mathbf{H}}_0 \bar{\mathbf{H}}_0^{\mathrm{T}} - \bar{\mathbf{h}}_0^{\mathrm{av}} (\bar{\mathbf{h}}_0^{\mathrm{av}})^{\mathrm{T}}, \tag{39b}$$

and

$$\mathbf{h}_0^{\mathrm{av}} := \frac{1}{N} \mathbf{H}_0 \mathbf{1}, \quad \mathbf{h}_1^{\mathrm{av}} := \frac{1}{N} \mathbf{H}_1 \mathbf{1}. \tag{40}$$

Note that the barred vectors and matrices denote the representation of their unbarred counterparts in the space of proper orthogonal modes. Accordingly,

$$\bar{\mathbf{h}}_0^{\mathrm{av}} := \mathbf{U}^{\mathrm{T}} \mathbf{h}_0^{\mathrm{av}}, \quad \bar{\mathbf{h}}_1^{\mathrm{av}} := \mathbf{U}^{\mathrm{T}} \mathbf{h}_1^{\mathrm{av}}, \quad \bar{\mathbf{H}}_0 := \mathbf{U}^{\mathrm{T}} \mathbf{H}_0, \quad \bar{\mathbf{H}}_1 := \mathbf{U}^{\mathrm{T}} \mathbf{H}_1. \tag{41}$$

Here, $\mathbf{h}_0^{\mathrm{av}}$ and $\mathbf{h}_1^{\mathrm{av}}$ are the average of concentration-deviation data in $\mathbf{H}_0$ and $\mathbf{H}_1$, respectively. Moreover, $\bar{\mathbf{X}}_{10}$ and $\bar{\mathbf{X}}_0$ are the time-delayed autocorrelation and covariance matrices of the time-series data represented with respect to the proper orthogonal modes. Note that equation 38a may be regarded as a generalization of the DMD solution (equation 33) it approaches to as $\omega \to \mathbf{0}$, which in turn occurs when the steady state has been attained. We also emphasize that metrics based on time-delayed autocorrelation were used in previous top-down approaches along with clustering algorithms to identify concentration pools irrespective of the underlying biochemical mechanisms (7, 8). However, as we highlighted in the previous section, what underlies the formation of concentration pools and coherent structures in biochemical reaction networks is the presence of timescale hierarchies. Furthermore, the optimal solution $\widehat{\mathbf{F}}$ in equation 38a implies that the time-delayed autocorrelation matrix contains information about the local Jacobian spectra. Although the expressions used in previous studies are not identical to equation 38a, our analysis demonstrate why time-delayed autocorrelation is a useful metric for statistical classifications of concentration pools.

Once $\widehat{\mathbf{F}}$ has been computed from equation 38a, the dominant eigenvalues and eigenmodes can readily be determined (Fig. 1E). Let $\bar{\boldsymbol{\Theta}} \boldsymbol{\Xi} \bar{\boldsymbol{\Theta}}^{-1}$ be the eigenvalue decomposition of $\widehat{\mathbf{F}}$ with $\boldsymbol{\Xi}$ a diagonal matrix, where the diagonal entries $\Xi_{ii} := \mu_i$ are the eigenvalues of $\widehat{\mathbf{F}}$ (not to be confused with the approximate eigenvalues of Toy Model 1 in the previous section). Then, the columns of $\boldsymbol{\Theta} := \mathbf{U}\bar{\boldsymbol{\Theta}} = [\theta_1 \cdots \theta_\nu]$ correspond to the dominant eigenmodes of $\mathbf{A}$ in the current time window. Because $\mathbf{A}$ and $\mathbf{F}$ are related through a similarity transformation, $\mu_i$ is also the dominant eigenvalues of $\mathbf{A}$. From equation 29, it follows that

$$\lambda_i = \frac{\log(\mu_i)}{\Delta t}, \quad i = 1 \cdot \cdot \nu, \tag{42}$$

with $\lambda_i$ the dominant eigenvalues of the local Jacobian matrix $\mathbf{J}$.

The next steps of DMA are similar to those of ODMD (20), but modifications are required due to the inhomogeneity of equation 34. Here, we note that the inhomogeneous term $\omega$ can be eliminated by introducing the differential time-series data matrix $\mathbf{D} := \mathbf{H}_1 - \mathbf{H}_0$, so that

$$\mathbf{d}_{k+1} = \mathbf{A}\mathbf{d}_k + \delta\epsilon_k, \quad k = 1 \cdot \cdot N, \tag{43}$$

where $\mathbf{d}_k := \mathbf{h}_{k+1} - \mathbf{h}_k$ is the $k$th column of $\mathbf{D}$ and $\delta\epsilon_k := \epsilon_{k+1} - \epsilon_k$. Since equation 43 is a homogenous system, we can follow the same procedure as ODMD to determine the optimal amplitudes associated with the dominant eigenvalues and eigenmodes ascertained in the previous step. Let

$$\mathbf{D} = \mathcal{U}\mathcal{S}\mathcal{V}^{\mathrm{T}}, \quad \mathcal{U} \in \mathbb{R}^{n \times \nu}, \quad \mathcal{S} \in \mathbb{R}^{\nu \times \nu}, \quad \mathcal{V} \in \mathbb{R}^{N \times \nu},$$

be the singular-value decomposition of $\mathbf{D} \in \mathbb{R}^{n \times N}$ and $\boldsymbol{\Gamma} \in \mathbb{C}^{\nu \times N}$ the Vandermonde matrix constructed from the eigenvalues $\{\mu_1, \cdots, \mu_\nu\}$. Then, the optimal amplitudes $\boldsymbol{\alpha}$ are given by (20)

$$\boldsymbol{\alpha} = \mathbf{L}^{-1}\mathbf{q}, \quad \mathbf{L} := \left(\bar{\boldsymbol{\Theta}}^* \bar{\boldsymbol{\Theta}}\right) \circ \overline{(\boldsymbol{\Gamma}\boldsymbol{\Gamma}^*)}, \quad \mathbf{q} := \overline{\mathrm{diag}\left(\boldsymbol{\Gamma}\mathcal{V}\mathcal{S}^{\mathrm{T}}\bar{\boldsymbol{\Theta}}\right)}, \tag{44}$$

where an overline denotes the complex conjugate of a matrix, the superscript $*$ denotes the complex-conjugate transpose, and $\circ$ indicates elementwise matrix multiplication. Accordingly, the modal matrix reads

$$\boldsymbol{\Phi} = \boldsymbol{\Theta}\boldsymbol{\Delta}, \tag{45}$$

the columns of which $\boldsymbol{\phi}_i$ are the dominant eigenmodes of $\mathbf{J}$. Here, $\boldsymbol{\Delta} \in \mathbb{C}^{\nu \times \nu}$ is a diagonal matrix with diagonal entries $\Delta_{ii} := \alpha_i$. Finally, we define the pooling matrix as

$$\mathbf{P} := \boldsymbol{\Phi}^{\dagger}. \tag{46}$$

Note that the pooling maps defined in the previous section correspond to the rows of $\mathbf{P}$. Here, the definition equation 46 ensures that the modal and pooling matrices satisfy the reciprocal orthogonality conditions in all time intervals. It is also consistent with the relaxation of the slowest eigenmode discussed in the previous section. Recall, we defined the pooling map near the steady state corresponding to the slowest mode as a normalized form of the slowest eigenmode, which is compatible with equation 46 since the Moore-Penrose inverse of a vector yields the transpose of the same vector with a normalized length.

The analysis of concentration pools and coherent structures for a given reaction system using the approach introduced in this section depend on the initial conditions and how they are perturbed. Because the results may vary based on the specific perturbations applied, dependence on initial conditions could be considered a limitation. However, we note that these concentration pools and coherent structures emerge from the inherent timescales of the system's dynamics. As a result, the characteristics of these structures should, in principle, remain consistent across different perturbations. Performing this analysis for a wide range of perturbations can furnish a more comprehensive and accurate description of concentration pools and coherent structures. Therefore, while the method may seem sensitive to initial conditions, it is not inherently limiting.

## Reciprocal orthogonality conditions for Toy Model 1

The physical interpretation of the reciprocal orthogonality conditions equations 12a and 12b is connected to the dissipative structure of chemical reaction networks. Since the transitory timescale $\check{T}_1$ that we are concerned with bridges two dynamic regimes associated with a fast ($T_1$) and slow ($T_2$) timescale characterizing the relaxation of reactions 2 and 3, their respective local eigenmodes represent the equilibration direction of these reactions in the concentration space. In particular, the slow eigenmode $\boldsymbol{\phi}_2$, which is the mode that $\check{\boldsymbol{\phi}}_1$ converges to when $t \to T_2$, aligns with equilibration directions of reaction 3:

$$\vartheta_3^{eq} = \mathbf{g}_3^{T} \boldsymbol{\chi}^{eq,3} = 0, \quad \mathbf{g}_3 := \left[ 0, k_3^+, -k_3^-, 0 \right]^{T}, \tag{47}$$

where $\boldsymbol{\chi}^{eq,3}$ denotes an equilibration direction. Concentration vectors that are parallel to this direction leave reaction 3 at its steady state, while those perpendicular to this direction maximally change the flux of reaction 3. Therefore, by requiring $\mathbf{p}_1$ to be orthogonal to $\boldsymbol{\phi}_2$ in equation 12a, we ensure that, at any time $t \in [T_1, T_2]$ along a dynamic trajectory, it points in a direction in the concentration space that maximizes the energy dissipation of reaction 3. In contrast, the fast eigenmode $\boldsymbol{\phi}_1$ aligns with the disequilibrium direction of reaction 2 (see chapter 4 of Palsson and Abrams [27]) for the definition of disequilibrium pools], where $\chi_1(t)$ and $\chi_2(t)$ are negatively correlated. Thus, by requiring $\check{\mathbf{p}}_1$ to be orthogonal to $\boldsymbol{\phi}_1$ in equation 12b, we ensure that it points in a direction corresponding to the conservation of metabolites involved in reaction 2 (see chapter 4 of Palsson and Abrams [(27] for the definition of conservation pools), where $\chi_1(t)$ and $\chi_2(t)$ are positively correlated.

The reciprocal orthogonality conditions in equations 12a and 12b also have an important geometric interpretation. For a given time interval along a dynamic trajectory, dominant eigenmodes represent the principal directions in a part of the concentration space where the dynamics occur. Therefore, if the underlying reaction system in the time interval of interest is low-dimensional, the eigenmodes can be viewed as a natural basis spanning a reduced concentration space where dynamic trajectories lie. Regarding this reduced concentration space as primal space (see Fig. 1F), equations 12a and 12b ensure that the vector representation of pools are a natural basis of a corresponding dual space. From this standpoint, the pools $p_1$ and $\check{p}_1$ in Toy Model 1 are scalar quantities defined as the inner product of a vector in the primal with another vector in the dual space, the algebraic form of which remain invariant with respect to any linear coordinate transformation (46). This geometric interpretation is compatible with the physical interpretation of flux-concentration duality in dissipative systems (43). As mentioned above, the vector representations of pools are closely related to flux-disturbance relaxation or maximal energy dissipation of reactions, so they form a basis for a dual space in which to represent flux dynamics naturally. Similarly, eigenmodes constitute a basis for a primal space in which to represent concentration dynamics naturally.

## Derivation of the third and fourth approximate eigenmodes of Toy Model 1

Here, we provide more details on the derivation of approximate expressions for the third and fourth eigenmodes of Toy Model 1 discussed in previous sections. Because the third eigenmode $\boldsymbol{\phi}_3$ arises from the second transitory eigenmode $\check{\boldsymbol{\phi}}_2$ in the interval $\left[ \check{T}_2, T_3 \right]$, we may regard $\boldsymbol{\phi}_3$ as the limit of $\check{\boldsymbol{\phi}}_2$ when $t \to T_3$. The second transitory regime occurs in the interval $\left[ \check{T}_2, T_3 \right]$ when reaction 3 equilibrates. In this transition period, the flux deviations of reactions 3 and 4 are of the same order, so that $\vartheta_2 \simeq \theta^a \vartheta_3$ and $\vartheta_3 \sim \vartheta_4 \gg \vartheta_1, \vartheta_5$. As with the first transitory period, we consider a coefficient $\beta$ such that $\vartheta_3 = \beta \vartheta_4$ to quantify the

transition between $T_2$ and $T_3$ with $\beta \sim \mathcal{O}(1)$. Accordingly, the mass-balance equations simplify to

$$\frac{d\chi_1}{dt} = -\theta^a\beta\vartheta_4, \tag{48a}$$

$$\frac{d\chi_2}{dt} = (\theta^a - 1)\beta\vartheta_4, \tag{48b}$$

$$\frac{d\chi_3}{dt} = (\beta - 1)\vartheta_4, \tag{48c}$$

$$\frac{d\chi_4}{dt} = \vartheta_4. \tag{48d}$$

Eliminating $\chi_3$ and $\chi_4$, equations 48a–48d is expressed with respect to $\vartheta_4$ as follows:

$$\frac{d\vartheta_4}{dt} = \left[k_4^+(\beta - 1) - k_4^-\right]\vartheta_4 \quad \Rightarrow \quad \vartheta_4(t) = \vartheta_4^0\exp(\check{\mu}_2 t),$$

where $\check{\mu}_2 := k_4^+(\beta - 1) - k_4^-$, and $\vartheta_4^0$ is the initial condition for the flux deviation of reaction 4. Substituting the solution of $\vartheta_4$ back in equations 48a–48d, we arrive at

$$\begin{bmatrix} \chi_1 \\ \chi_2 \\ \chi_3 \\ \chi_4 \end{bmatrix} = \vartheta_4^0/\check{\mu}_2 \underbrace{\begin{bmatrix} -\theta^a\beta \\ (\theta^a - 1)\beta \\ \beta - 1 \\ 1 \end{bmatrix}}_{\check{\phi}_2}\exp(\check{\mu}_2 t) + \begin{bmatrix} c_1 \\ c_2 \\ c_3 \\ c_4 \end{bmatrix}, \tag{49}$$

where $c_i$ are the integration constants. Note that the coefficient $\beta$ varies from a large value to its asymptotic value $\beta^a$ as $t \to T_3$. This asymptotic coefficient is in turn determined from the equilibrium of Reaction 3

$$\vartheta_3 \to 0 \quad \text{as} \quad t \to T_3 \quad \Rightarrow \quad \chi_3 \to K_3^{eq}\chi_2 \quad \text{and} \quad \frac{d\chi_3}{dt} \to K_3^{eq}\frac{d\chi_2}{dt}, \tag{50}$$

with $K_3^{eq} := k_3^+/k_3^-$ the equilibrium constant of reaction 3. It follows from equations 50, 48b, and 48c that

$$\beta^a = \frac{K_2^{eq} + 1}{K_2^{eq}K_3^{eq} + K_2^{eq} + 1}. \tag{51}$$

We now can express $\phi_3$ and $\mu_3$ as the limiting cases of $\check{\phi}_2$ and $\check{\mu}_2$:

$$\phi_3 = \lim_{t \to T_3} \check{\phi}_2 = \begin{bmatrix} -\theta^a\beta^a \\ (\theta^a - 1)\beta^a \\ \beta^a - 1 \\ 1 \end{bmatrix}, \quad \mu_3 = \lim_{t \to T_3} \check{\mu}_2 = k_4^+(\beta^a - 1) - k_4^-. \tag{52}$$

The third transitory regime occurs in the interval $\left[\check{T}_3, T_\infty\right]$, where $T_\infty$ is a sufficiently large time—referred to as the total relaxation time—by which all flux and concentration disturbances have relaxed to within a tolerable threshold. This regime is characterized by $\vartheta_2 \simeq \theta^a\vartheta_3$, $\vartheta_3 \simeq \beta^a\vartheta_4$, and $\vartheta_4 \sim \vartheta_1, \vartheta_5$. We express the order-of-magnitude balance between the flux deviations of reactions 4 and 5 by a coefficient $\gamma$ such that $\vartheta_4 = \gamma\vartheta_5$

to describe the transitory regime between $T_3$ and $T_\infty$ with $\gamma \sim \mathcal{O}(1)$, simplifying the mass-balance equations to

$$\frac{d\chi_1}{dt} = \vartheta_1 - \theta^a \beta^a \gamma \vartheta_5, \tag{53a}$$

$$\frac{d\chi_2}{dt} = (\theta^a - 1)\beta^a \gamma \vartheta_5, \tag{53b}$$

$$\frac{d\chi_3}{dt} = (\beta^a - 1)\gamma \vartheta_5, \tag{53c}$$

$$\frac{d\chi_4}{dt} = (\gamma - 1)\vartheta_5. \tag{53d}$$

From equation 53d, it follows that

$$\frac{d\vartheta_5}{dt} = k_5^+(\gamma - 1)\vartheta_5 \quad \Rightarrow \quad \vartheta_5(t) = \vartheta_5^0 \exp(\check{\mu}_3 t),$$

where $\check{\mu}_3 := k_5^+(\gamma - 1)$, and $\vartheta_5^0$ is the initial condition for the flux deviation of reaction 5. Substituting $\vartheta_5(t)$ from this equation in equation 53a, we obtain the following solution for $\vartheta_1(t)$:

$$\vartheta_1(t) = \frac{k_1^- \theta^a \beta^a \gamma \vartheta_5^0}{\check{\mu}_3 + k_1^-}[\exp(\check{\mu}_3 t) - \exp(-k_1^- t)] + \vartheta_1^0 \exp(-k_1^- t),$$

with $\vartheta_1^0$ the initial condition for the flux deviation of reaction 1. From this solution, it follows that the dynamics of $\vartheta_1$ in the third transitory regime are described by two exponential decay modes of the same order of magnitude (note that $k_5^+ \sim k_1^-$ in Toy Model 1; see Fig. S1C). Our goal in the analysis of this transitory regime is to ascertain the slowest eigenmode of Toy Model 1 and describe the dynamics near its steady state. Therefore, we assume $|\check{\mu}_3| < k_1^-$ in the remainder of this section, so that the slowest timescale is characterized by $\check{\mu}_3$. A similar analysis can be performed for the opposite case. Substituting the solutions of $\vartheta_1$ and $\vartheta_5$ in equations 53a–53d and neglecting the faster mode associated with $\exp(-k_1^- t)$, we obtain the dynamic trajectory of concentration deviations:

$$\begin{bmatrix} \chi_1 \\ \chi_2 \\ \chi_3 \\ \chi_4 \end{bmatrix} = \vartheta_5^0/\check{\mu}_3 \underbrace{\begin{bmatrix} \dfrac{\kappa(1-\gamma)}{\kappa(\gamma-1)+1}\theta^a \beta^a \gamma \\ (\theta^a - 1)\beta^a \gamma \\ (\beta^a - 1)\gamma \\ \gamma - 1 \end{bmatrix}}_{\check{\phi}_3} \exp(\check{\mu}_3 t) + \begin{bmatrix} c_1 \\ c_2 \\ c_3 \\ c_4 \end{bmatrix}, \tag{54}$$

where $\kappa := k_5^+/k_1^-$. Since the flux deviations of reaction 1 becomes nonnegligible in this regime, the equilibration of reactions 2 and 3 is disturbed once again on the third transitory timescale $\check{T}_3$. Therefore, the asymptotic coefficients $\theta^a$ and $\beta^a$ change during the secondary relaxation of reactions 2 and 3 as $t \to T_\infty$. We determine the asymptotic value of $\theta$ in the third transitory regime from equations 6, 53a, and 53b:

$$\theta^a = \frac{\kappa(\gamma^a - 1) + 1}{\kappa(\gamma^a - 1)(K_2^{eq} + 1) + 1}, \tag{55}$$

and of $\beta$ from equations 50, 53b, and 53c:

$$\beta^{a} = \frac{\kappa(\gamma^{a} - 1)(K_2^{eq} + 1) + 1}{\kappa(\gamma^{a} - 1)(K_2^{eq} K_3^{eq} + K_2^{eq} + 1) + 1}. \tag{56}$$

The coefficient $\gamma$ varies from a large value to its asymptotic value $\gamma^{a}$ as $t \to T_{\infty}$, which is ascertained from the equilibrium of reaction 4:

$$\vartheta_4 \to 0 \quad \text{as} \quad t \to T_{\infty} \quad \Rightarrow \quad \chi_4 \to K_4^{eq} \chi_3 \quad \text{and} \quad \frac{d\chi_4}{dt} \to K_4^{eq} \frac{d\chi_3}{dt}, \tag{57}$$

where $K_4^{eq} := k_4^+ / k_4^-$ is the equilibrium constant of reaction 4. From equations 57, 53c, and 53d, we find that

$$\gamma^{a} + \frac{\kappa \gamma^{a}(\gamma^{a} - 1) K_2^{eq} K_3^{eq} K_4^{eq}}{\kappa(\gamma^{a} - 1)(K_2^{eq} K_3^{eq} + K_2^{eq} + 1) + 1} = 1, \tag{58}$$

which is a quadratic polynomial to be solved with respect to $\gamma^{a}$. As with previous cases, $\boldsymbol{\phi}_4$ and $\mu_4$ are the limiting cases of $\check{\boldsymbol{\phi}}_3$ and $\check{\mu}_3$, so that

$$\boldsymbol{\phi}_4 = \lim_{t \to T_{\infty}} \check{\boldsymbol{\phi}}_3 = \begin{bmatrix} \dfrac{\kappa(1 - \gamma^{a})}{\kappa(\gamma^{a} - 1) + 1} \theta^{a} \beta^{a} \gamma^{a} \\ (\theta^{a} - 1)\beta^{a}\gamma^{a} \\ (\beta^{a} - 1)\gamma^{a} \\ \gamma^{a} - 1 \end{bmatrix}, \quad \mu_4 = \lim_{t \to T_{\infty}} \check{\mu}_3 = k_5^+(\gamma^{a} - 1). \tag{59}$$

## Inhomogeneity of evolution equations for local concentration deviations and temporal grid size

We show that the discrete form of evolution equations for local concentration deviations (equation 34) becomes homogeneous asymptotically for vanishingly small temporal grid size, that is, $\boldsymbol{\omega} \to \mathbf{0}$ as $\Delta t \to 0$. We start from the definition of $\boldsymbol{\omega}$ in equation 34:

$$\boldsymbol{\omega} := \mathbf{A}\mathbf{w} - \mathbf{w}.$$

Substituting the definitions of $\mathbf{A}$ and $\mathbf{w}$ from previous sections (see "Dynamic mode analysis") in this equation, we have

$$\boldsymbol{\omega} = \text{Exp}(\mathbf{J}\Delta t)\mathbf{J}^{-1}\boldsymbol{a} - \mathbf{J}^{-1}\boldsymbol{a} \tag{60a}$$

$$= \left(\mathbf{I} + \mathbf{J}\Delta t + \frac{1}{2!}\mathbf{J}^2\Delta t^2 + \cdots\right)\mathbf{J}^{-1}\boldsymbol{a} - \mathbf{J}^{-1}\boldsymbol{a} \tag{60b}$$

$$= \mathbf{J}^{-1}\boldsymbol{a} + \mathbf{I}\boldsymbol{a}\Delta t + \frac{1}{2!}\mathbf{J}\boldsymbol{a}\Delta t^2 + \cdots - \mathbf{J}^{-1}\boldsymbol{a} \tag{60c}$$

$$= \left(\mathbf{I} + \frac{1}{2!}\mathbf{J}\Delta t + \frac{1}{3!}\mathbf{J}^2\Delta t^2 + \cdots\right)\boldsymbol{a}\Delta t. \tag{60d}$$

In equation 60d, the expression enclosed in parentheses is a matrix with a nonzero norm. Moreover, the constant vector $\boldsymbol{a}$ is generally nonzero in all sliding time windows, and it only approaches zero asymptotically as dynamic trajectories near a steady-state solution. It follows that the inhomogeneous term $\boldsymbol{\omega}$ approaches zero asymptotically in time windows away from the steady state only if an arbitrarily small temporal grid size $\Delta t$ is chosen.

## Relationship between time-delayed autocorrelation, covariance, and local Jacobian spectra

Here, we prove that $\widehat{\mathbf{F}}$ and $\widehat{\omega}$ in equations 38a–38b are the solution of the optimization problem (equation 37). We begin by expanding the objective:

$$\|\boldsymbol{\mathcal{E}}\|_F^2 = \underbrace{\mathrm{tr}\big(\bar{\mathbf{H}}_0^\mathrm{T}\mathbf{F}^\mathrm{T}\mathbf{F}\bar{\mathbf{H}}_0\big)}_{I} + \underbrace{\mathrm{tr}\big(\bar{\mathbf{W}}^\mathrm{T}\bar{\mathbf{W}}\big)}_{II} + \underbrace{2\mathrm{tr}\big(\bar{\mathbf{H}}_0^\mathrm{T}\mathbf{F}^\mathrm{T}\bar{\mathbf{W}}\big)}_{III}$$
$$- \underbrace{2\mathrm{tr}\big(\bar{\mathbf{H}}_0^\mathrm{T}\mathbf{F}^\mathrm{T}\bar{\mathbf{H}}_1\big)}_{IV} - \underbrace{2\mathrm{tr}\big(\bar{\mathbf{H}}_1^\mathrm{T}\bar{\mathbf{W}}\big)}_{V} + \cdots, \tag{61}$$

where $\bar{\mathbf{W}} := \mathbf{U}^\mathrm{T}\mathbf{W}$, and the dots represent constant terms that do not affect the optimal solution of equation 37. As stated before, $\mathbf{U}$ can be regarded as a transformation, mapping matrices and vectors from the space of proper orthogonal modes into the original concentration space, so we may write $\bar{\mathbf{W}} = \bar{\omega}\mathbf{1}^\mathrm{T}$ with $\bar{\omega} := \mathbf{U}^\mathrm{T}\omega$. In the following, we further expand each term in equation 61 to derive expressions that can be readily recast into a standard-form quadratic program.

We start from the first term

$$I = \mathrm{tr}\big(\mathbf{F}\bar{\mathbf{H}}_0\bar{\mathbf{H}}_0^\mathrm{T}\mathbf{F}^\mathrm{T}\big) \tag{62a}$$
$$= \mathrm{vec}\big(\bar{\mathbf{H}}_0^\mathrm{T}\mathbf{F}^\mathrm{T}\big)^\mathrm{T}\mathrm{vec}\big(\bar{\mathbf{H}}_0^\mathrm{T}\mathbf{F}^\mathrm{T}\big) \tag{62b}$$
$$= \big[\big(\mathbf{I}_\nu \otimes \bar{\mathbf{H}}_0^\mathrm{T}\big)\mathrm{vec}\big(\mathbf{F}^\mathrm{T}\big)\big]^\mathrm{T}\big(\mathbf{I}_\nu \otimes \bar{\mathbf{H}}_0^\mathrm{T}\big)\mathrm{vec}\big(\mathbf{F}^\mathrm{T}\big) \tag{62c}$$
$$= \mathrm{vec}\big(\mathbf{F}^\mathrm{T}\big)^\mathrm{T}\big(\mathbf{I}_\nu \otimes \bar{\mathbf{H}}_0^\mathrm{T}\big)^\mathrm{T}\big(\mathbf{I}_\nu \otimes \bar{\mathbf{H}}_0^\mathrm{T}\big)\mathrm{vec}\big(\mathbf{F}^\mathrm{T}\big) \tag{62d}$$
$$= \mathrm{vec}\big(\mathbf{F}^\mathrm{T}\big)^\mathrm{T}\big(\mathbf{I}_\nu \otimes \bar{\mathbf{H}}_0\big)\big(\mathbf{I}_\nu \otimes \bar{\mathbf{H}}_0^\mathrm{T}\big)\mathrm{vec}\big(\mathbf{F}^\mathrm{T}\big) \tag{62e}$$
$$= \mathrm{vec}\big(\mathbf{F}^\mathrm{T}\big)^\mathrm{T}\big(\mathbf{I}_\nu \otimes \bar{\mathbf{H}}_0\bar{\mathbf{H}}_0^\mathrm{T}\big)\mathrm{vec}\big(\mathbf{F}^\mathrm{T}\big), \tag{62f}$$

where $\mathbf{I}_\nu$ denotes a $\nu \times \nu$ identity matrix, and $\mathrm{vec}(\cdot)$ vectorizes a matrix by stacking its columns on top of one another. Here, we used the cyclic property of the trace in equation 62a and applied the identity $\mathrm{vec}\big(\bar{\mathbf{H}}_0^\mathrm{T}\mathbf{F}^\mathrm{T}\big) = \big(\mathbf{I}_\nu \otimes \bar{\mathbf{H}}_0^\mathrm{T}\big)\mathrm{vec}\big(\mathbf{F}^\mathrm{T}\big)$ in equation 62c. Expanding the second term, we arrive at

$$II = \mathrm{vec}\big(\bar{\mathbf{W}}\big)^\mathrm{T}\mathrm{vec}\big(\bar{\mathbf{W}}\big) = N\bar{\omega}^\mathrm{T}\bar{\omega}. \tag{63}$$

The third term is written as follows:

$$III = \mathrm{tr}\big(\bar{\mathbf{W}}\bar{\mathbf{H}}_0^\mathrm{T}\mathbf{F}^\mathrm{T}\big) \tag{64a}$$
$$= \mathrm{vec}\big(\bar{\mathbf{W}}^\mathrm{T}\big)^\mathrm{T}\mathrm{vec}\big(\bar{\mathbf{H}}_0^\mathrm{T}\mathbf{F}^\mathrm{T}\big) \tag{64b}$$
$$= \big(\bar{\omega} \otimes \mathbf{1}\big)^\mathrm{T}\big(\mathbf{I}_\nu \otimes \bar{\mathbf{H}}_0^\mathrm{T}\big)\mathrm{vec}\big(\mathbf{F}^\mathrm{T}\big) \tag{64c}$$
$$= \big(\bar{\omega}^\mathrm{T} \otimes \mathbf{1}^\mathrm{T}\big)\big(\mathbf{I}_\nu \otimes \bar{\mathbf{H}}_0^\mathrm{T}\big)\mathrm{vec}\big(\mathbf{F}^\mathrm{T}\big) \tag{64d}$$
$$= \big(\bar{\omega}^\mathrm{T} \otimes \mathbf{1}^\mathrm{T}\bar{\mathbf{H}}_0^\mathrm{T}\big)\mathrm{vec}\big(\mathbf{F}^\mathrm{T}\big) \tag{64e}$$
$$= \big[\bar{\omega}^\mathrm{T} \otimes N(\bar{\mathbf{h}}_0^\mathrm{av})^\mathrm{T}\big]\mathrm{vec}\big(\mathbf{F}^\mathrm{T}\big) = N\sum_{i=1}^{\nu} \bar{\omega}_i(\bar{\mathbf{h}}_0^\mathrm{av})^\mathrm{T}f_i, \tag{64f}$$

where $f_i$ is the $i$th column of $\mathbf{F}^\mathrm{T}$. Again, we used the cyclic property of the trace in equation 64a and the identity $\mathrm{vec}\big(\bar{\mathbf{W}}^\mathrm{T}\big) = \bar{\omega} \otimes \mathbf{1}$ in equation 64c. Expanding the fourth term, we have

$$IV = \text{tr}\big(\bar{\mathbf{H}}_1\bar{\mathbf{H}}_0^{\mathrm{T}}\mathbf{F}^{\mathrm{T}}\big) \tag{65a}$$

$$= \text{vec}\big(\bar{\mathbf{H}}_0\bar{\mathbf{H}}_1^{\mathrm{T}}\big)^{\mathrm{T}}\text{vec}\big(\mathbf{F}^{\mathrm{T}}\big) \tag{65b}$$

$$= \Big[\big(\mathbf{I}_\nu \otimes \bar{\mathbf{H}}_0\big)\text{vec}\big(\bar{\mathbf{H}}_1^{\mathrm{T}}\big)\Big]^{\mathrm{T}}\text{vec}\big(\mathbf{F}^{\mathrm{T}}\big) = \bar{\boldsymbol{\beta}}^{\mathrm{T}}\text{vec}\big(\mathbf{F}^{\mathrm{T}}\big), \tag{65c}$$

where

$$\bar{\boldsymbol{\beta}} := \big(\mathbf{I}_\nu \otimes \bar{\mathbf{H}}_0\big)\text{vec}\big(\bar{\mathbf{H}}_1^{\mathrm{T}}\big) = \begin{bmatrix} \bar{\mathbf{H}}_0\bar{\boldsymbol{h}}_{1,1} \\ \bar{\mathbf{H}}_0\bar{\boldsymbol{h}}_{1,2} \\ \vdots \\ \bar{\mathbf{H}}_0\bar{\boldsymbol{h}}_{1,\nu} \end{bmatrix} \tag{66}$$

with $\bar{\boldsymbol{h}}_{1,i}$ denoting the $i$th column of $\bar{\mathbf{H}}_1^{\mathrm{T}}$. The fifth term is expanded similarly:

$$V = \text{vec}\big(\bar{\mathbf{H}}_1\big)^{\mathrm{T}}\text{vec}\big(\bar{\mathbf{W}}\big) = \left(\sum_{i=1}^{N} \bar{\mathbf{h}}_{1,i}^{\mathrm{T}}\right)\bar{\boldsymbol{\omega}} = N(\bar{\mathbf{h}}_1^{\mathrm{av}})^{\mathrm{T}}\bar{\boldsymbol{\omega}} = \bar{\boldsymbol{\rho}}^{\mathrm{T}}\bar{\boldsymbol{\omega}}, \tag{67}$$

where $\bar{\mathbf{h}}_{1,i}$ denotes the $i$th column of $\bar{\mathbf{H}}_1$, and $\bar{\boldsymbol{\rho}} := N\bar{\mathbf{h}}_1^{\mathrm{av}}$.

We now can rewrite the optimization problem equation 37 as a quadratic program in standard form by substituting the five terms $I$–$V$ from equations 62f, 63, 64f, 65c, 66, and 67 in equation 61:

$$\hat{\mathbf{z}} := \arg\min_{\mathbf{z}} \ \mathbf{z}^{\mathrm{T}}\mathcal{H}\mathbf{z} - 2b^{\mathrm{T}}\mathbf{z}. \tag{68}$$

Here,

$$\mathcal{H} := \begin{bmatrix} \mathcal{A} & \mathcal{B} \\ \hline \mathcal{C} & \mathcal{D} \end{bmatrix}, \quad b := \begin{bmatrix} \bar{\boldsymbol{\beta}} \\ \bar{\boldsymbol{\rho}} \end{bmatrix}, \quad \mathbf{z} := \begin{bmatrix} \text{vec}\big(\mathbf{F}^{\mathrm{T}}\big) \\ \bar{\boldsymbol{\omega}} \end{bmatrix} \tag{69}$$

with

$$\mathcal{A} := \overbrace{\begin{bmatrix} \bar{\mathbf{H}}_0\bar{\mathbf{H}}_0^{\mathrm{T}} & \cdots & \mathbf{0} \\ \vdots & \ddots & \vdots \\ \mathbf{0} & \cdots & \bar{\mathbf{H}}_0\bar{\mathbf{H}}_0^{\mathrm{T}} \end{bmatrix}}^{\nu\ \text{blocks}} \in \mathbb{R}^{\nu^2 \times \nu^2}, \quad \mathbf{B} := \overbrace{\begin{bmatrix} \bar{\mathbf{h}}_0^{\mathrm{av}} & \cdots & \mathbf{0} \\ \vdots & \ddots & \vdots \\ \mathbf{0} & \cdots & \bar{\mathbf{h}}_0^{\mathrm{av}} \end{bmatrix}}^{\nu\ \text{blocks}} \in \mathbb{R}^{\nu \times \nu^2}, \tag{70}$$

$\mathcal{B} := N\mathbf{B}^{\mathrm{T}}$, $\mathcal{C} := N\mathbf{B}$, and $\mathcal{D} := N\mathbf{I}_\nu$. The solution of the minimization problem (equation 68) is ascertained from the Karush-Kuhn-Tucker conditions (47):

$$\hat{\mathbf{z}} = \mathcal{H}^{-1}b. \tag{71}$$

To derive compact from expressions for $\widehat{\mathbf{F}}$ and $\hat{\boldsymbol{\omega}}$, the Hessian $\mathcal{H}$ is to be inverted explicitly with respect to its elements. Since the Hessian in equation 69 is a block matrix, we derive its block inverse

$$\mathcal{H}^{-1} = \begin{bmatrix} \mathcal{A}^{\#} & \mathcal{B}^{\#} \\ \hline \mathcal{C}^{\#} & \mathcal{D}^{\#} \end{bmatrix} \tag{72}$$

using the Schur complement. With respect to the matrix blocks in equation 69, the Schur complement of $\mathcal{H}$ is written as follows:

$$\tilde{\mathcal{H}} := \mathcal{A} - \mathcal{B}\mathcal{D}^{-1}\mathcal{C} = \mathcal{A} - N\mathbf{B}^T\mathbf{B} \overbrace{\begin{bmatrix} \bar{\mathbf{X}}_0 & \cdots & \mathbf{0} \\ \vdots & \ddots & \vdots \\ \mathbf{0} & \cdots & \bar{\mathbf{X}}_0 \end{bmatrix}}^{\nu\ \text{blocks}}, \tag{73}$$

where $\bar{\mathbf{X}}_0$ is the covariance of time-series data $\bar{\mathbf{H}}_0$ defined in equation 39b. We can now express the individual blocks in equation 72 in terms of the Schur complement $\tilde{\mathcal{H}}$:

$$\mathcal{A}^{\#} = \tilde{\mathcal{H}}^{-1} = \frac{1}{N}\begin{bmatrix} \bar{\mathbf{X}}_0^{-1} & \cdots & \mathbf{0} \\ \vdots & \ddots & \vdots \\ \mathbf{0} & \cdots & \bar{\mathbf{X}}_0^{-1} \end{bmatrix}, \tag{74a}$$

$$\mathcal{B}^{\#} = -\tilde{\mathcal{H}}^{-1}\mathcal{B}\mathcal{D}^{-1} = -\frac{1}{N}\begin{bmatrix} \bar{\mathbf{X}}_0^{-1}\bar{\mathbf{h}}_0^{av} & \cdots & \mathbf{0} \\ \vdots & \ddots & \vdots \\ \mathbf{0} & \cdots & \bar{\mathbf{X}}_0^{-1}\bar{\mathbf{h}}_0^{av} \end{bmatrix}, \tag{74b}$$

$$\mathcal{C}^{\#} = -\mathcal{D}^{-1}\mathcal{C}\tilde{\mathcal{H}}^{-1} = -\frac{1}{N}\begin{bmatrix} (\bar{\mathbf{h}}_0^{av})^T\bar{\mathbf{X}}_0^{-1} & \cdots & \mathbf{0} \\ \vdots & \ddots & \vdots \\ \mathbf{0} & \cdots & (\bar{\mathbf{h}}_0^{av})^T\bar{\mathbf{X}}_0^{-1} \end{bmatrix}, \tag{74c}$$

$$\begin{aligned} \mathcal{D}^{\#} &= \mathcal{D}^{-1} + \mathcal{D}^{-1}\mathcal{C}\tilde{\mathcal{H}}^{-1}\mathcal{B}\mathcal{D}^{-1} \\ &= \frac{1}{N}\begin{bmatrix} 1 + (\bar{\mathbf{h}}_0^{av})^T\bar{\mathbf{X}}_0^{-1}\bar{\mathbf{h}}_0^{av} & \cdots & \mathbf{0} \\ \vdots & \ddots & \vdots \\ \mathbf{0} & \cdots & 1 + (\bar{\mathbf{h}}_0^{av})^T\bar{\mathbf{X}}_0^{-1}\bar{\mathbf{h}}_0^{av} \end{bmatrix}. \end{aligned} \tag{74d}$$

Accordingly, the two parts of the optimal solution $\hat{\mathbf{z}}$ are ascertained separately by plugging $\mathcal{H}^{-1}$ from equation 72 into equation 71:

$$\text{vec}\left(\widehat{\mathbf{F}}^T\right) = \mathcal{A}^{\#}\bar{\boldsymbol{\beta}} + \mathcal{B}^{\#}\bar{\boldsymbol{\rho}}, \tag{75a}$$

$$\widehat{\bar{\boldsymbol{\omega}}} = \mathcal{C}^{\#}\bar{\boldsymbol{\beta}} + \mathcal{D}^{\#}\bar{\boldsymbol{\rho}}. \tag{75b}$$

Substituting the inverse blocks from equations 74a and 74b in equation 75a, we obtain the first part

$$\text{vec}\left(\widehat{\mathbf{F}}^T\right) = \begin{bmatrix} \bar{\mathbf{X}}_0^{-1}\left(\bar{\mathbf{H}}_0\bar{\boldsymbol{h}}_{1,1}/N - \bar{\mathbf{h}}_0^{av}\bar{h}_{1,1}^{av}\right) \\ \bar{\mathbf{X}}_0^{-1}\left(\bar{\mathbf{H}}_0\bar{\boldsymbol{h}}_{1,2}/N - \bar{\mathbf{h}}_0^{av}\bar{h}_{1,2}^{av}\right) \\ \vdots \\ \bar{\mathbf{X}}_0^{-1}\left(\bar{\mathbf{H}}_0\bar{\boldsymbol{h}}_{1,\nu}/N - \bar{\mathbf{h}}_0^{av}\bar{h}_{1,\nu}^{av}\right) \end{bmatrix} \tag{76}$$

with $\bar{h}_{1,i}^{av}$ denoting the $i$th component of $\bar{\mathbf{h}}_1^{av}$. From equation 76, one can verify that the optimal solution in matrix form is

$$\widehat{\mathbf{F}}^T = \bar{\mathbf{X}}_0^{-1}\bar{\mathbf{X}}_{01}, \tag{77}$$

where

$$\bar{\mathbf{X}}_{01} := \frac{1}{N}\bar{\mathbf{H}}_0\bar{\mathbf{H}}_1^T - \bar{\mathbf{h}}_0^{av}\left(\bar{\mathbf{h}}_1^{av}\right)^T. \tag{78}$$

Since $\bar{\mathbf{X}}_0$ is symmetric and $\bar{\mathbf{X}}_{10} = \bar{\mathbf{X}}_{01}^T$, the optimal solution $\widehat{\mathbf{F}}$ given in equation 38a follows from equation 77. Similarly, the second part of the optimal solution is obtained by substituting the inverse blocks from equations 74c and 74d in equation 75b:

$$\widehat{\bar{\boldsymbol{\omega}}} = \bar{\mathbf{h}}_1^{\mathrm{av}} - \widehat{\mathbf{F}}\bar{\mathbf{h}}_0^{\mathrm{av}}, \tag{79}$$

from which the optimal solution $\widehat{\boldsymbol{\omega}}$ given in equation 38b is derived using the relationship $\widehat{\boldsymbol{\omega}} = \mathbf{U}\widehat{\bar{\boldsymbol{\omega}}}$.

## Energetics of glycolysis

Overall, glycolysis breaks glucose down into pyruvate and lactate according to

$$\text{glucose} + 2\text{ADP} + 2\text{Pi} + \text{NAD} \rightarrow \text{pyruvate} + \text{lactate} + 2\text{ATP} + 2\text{H}_2\text{O} + \text{NADH} + \text{H}^+.$$

Since glucose carries a higher chemical potential energy than pyruvate and lactate combined, as measured by their relative transformed Gibbs free energy of formation $\Delta_f G'^\circ$ (48), the transformation glucose $\rightarrow$ pyruvate + lactate is exergonic:

$$\Delta_r G'^\circ = \Delta_f G'^\circ_{\mathrm{pyruvate}} + \Delta_f G'^\circ_{\mathrm{lactate}} - \Delta_f G'^\circ_{\mathrm{glucose}} \approx -241 \text{ kJ/mol} \quad \text{at} \quad \text{pH} = 7.5,$$

where $\Delta_r G'^\circ$ is the standard transformed Gibbs free energy of reaction. This energy drives the synthesis of high-energy cofactors (ATP and NADH) from the low-energy counterparts (ADP, Pi, and NAD) through interconversion reactions

$$\begin{aligned}
\text{ADP} + \text{Pi} \rightarrow \text{ATP} + \text{H}_2\text{O} \quad & \Delta_r G'^\circ \approx 30 \text{ kJ/mol} \quad \text{at} \quad \text{pH} = 7.5, \\
\text{NAD} \rightarrow \text{NADH} \quad & \Delta_r G'^\circ \approx 66 \text{ kJ/mol} \quad \text{at} \quad \text{pH} = 7.5.
\end{aligned}$$

Accordingly, ATP and NADH carry a higher chemical potential energy than ADP + Pi and NAD in an aqueous environment, respectively. In this sense, ATP and NADH are regarded as high-energy cofactors with ADP, Pi, and NAD the corresponding low-energy counterparts.

## ACKNOWLEDGMENTS

This work was funded by the Novo Nordisk Foundation (grant number NNF10CC1016517) and the National Institutes of Health (grant number GM057089).

Conceptualization: A.A. and B.O.P.; methodology: A.A.; validation: A.A.; formal analysis: A.A.; investigation: A.A. and Z.B.H.; writing – original draft: A.A.; writing – review & editing: A.A.: Z.B.H., and B.O.P.; funding acquisition: B.O.P.; resources: B.O.P.; and supervision: B.O.P.

## AUTHOR AFFILIATIONS

[1]Department of Bioengineering, University of California San Diego, La Jolla, California, USA

[2]Novo Nordisk Foundation Center for Biosustainability, Technical University of Denmark, Lyngby, Denmark

## AUTHOR ORCIDs

Amir Akbari http://orcid.org/0000-0002-4826-078X
Bernhard O. Palsson http://orcid.org/0000-0003-2357-6785

## FUNDING

| Funder | Grant(s) | Author(s) |
|---|---|---|
| Novo Nordisk Fonden (NNF) | NNF10CC1016517 | Bernhard O. Palsson |
| HHS \| National Institutes of Health (NIH) | GM057089 | Bernhard O. Palsson |

## DATA AVAILABILITY

All data generated or analyzed during this study are included in this published article and its supplemental files. The source code used in this study is available online (https://github.com/akbari84/Dynamic-Mode-Analysis.git).

## ADDITIONAL FILES

The following material is available online.

### Supplemental Material

**Data S1 (mSystems01001-23-s0001.xlsx).** Information about the kinetic model of the glycolytic pathway discussed in the text.
**Data S2 (mSystems01001-23-s0002.xlsx).** Complete pooling matrices of the glycolytic pathway associated with the sliding time window as it moves from the initial time to the steady state.
**Supplemental figures (mSystems01001-23-s0003.pdf).** Fig. S1-S4.

### Open Peer Review

**PEER REVIEW HISTORY (review-history.pdf).** An accounting of the reviewer comments and feedback.

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
