## [Reviewer comments · mSystems]

A data-driven approach for timescale decomposition of biochemical reaction networks

Amir Akbari, Zachary Haiman, and Bernhard Palsson

Corresponding Author(s): Bernhard Palsson, University of California San Diego Jacobs School of Engineering

Review Timeline:

Submission Date:	September 20, 2023
Editorial Decision:	October 16, 2023
Revision Received:	December 1, 2023
Accepted:	December 5, 2023

Editor: Lennart Schada von Borzyskowski

Reviewer(s): The reviewers have opted to remain anonymous.

Transaction Report:

DOI: <https://doi.org/10.1128/msystems.01001-23>

October 16, 2023

Dr. Bernhard O Palsson
University of California San Diego
La Jolla

Re: mSystems01001-23 (A data-driven approach for timescale decomposition of biochemical reaction networks)

Dear Dr. Bernhard O Palsson:

Thank you for submitting your manuscript to mSystems. We have completed our review and I am pleased to inform you that, in principle, we expect to accept it for publication in mSystems. However, acceptance will not be final until you have adequately addressed the reviewer comments.

Please make sure to address all reviewer comments adequately and alter/improve the manuscript text and figures when required.

Please also ensure to take care of the code availability, as pointed out by reviewer 2.

I am looking forward to receiving your replies to the reviewer comments as well as a revised version of the manuscript.

Preparing Revision Guidelines

Please return the manuscript within 60 days; if you cannot complete the modification within this time period, please contact me. If you do not wish to modify the manuscript and prefer to submit it to another journal, please notify me of your decision immediately so that the manuscript may be formally withdrawn from consideration by mSystems.

Sincerely,

Lennart Schada von Borzyskowski

Editor, mSystems

Journals Department
Reviewer comments:

Reviewer #1 (Comments for the Author):

Akbari, Haiman, and Palsson describe a method for calculating subsets (or pools) metabolites according to enzyme kinetics of reaction pathways in the manuscript entitled, 'A data-driven approach for timescale decomposition of biochemical reaction networks'. The Palsson group has published numerous papers on this topic over the past years, including textbooks, so the content has largely been described previously for metabolic and signaling networks. The novel contribution being added here is "Dynamic Model Analysis" for calculating dynamic pools of metabolites that is closely related to autocorrelation analysis. The present manuscript is lucid, descriptively systematic, and well-written.

There are a few points that may be considered in revising the manuscript,

The presented approach essentially involves linearization following a perturbation and then application of a sliding window. Based on this, it would appear that the definition of the pools are all dependent on the type(s) of perturbation that is(are) applied. This limits the generalizability of the approach and should probably be discussed.

It is not clear how the tolerance threshold (11a, 11b, Line 335) is selected. This appears to be a somewhat arbitrary parameter that can have significant impact on the ultimate calculation of the pooling matrix.

The dynamics for the glycolytic example (particularly for the first 5 pools) seem very slow in terms of what is typically expected for small molecule metabolism (see for example, cited reference #9, Kauffman et al). It is not clear if this is due to the model that was used versus a limitation of the method. For example, a statement is made with regards to consistency with circulation times (Line 159), however the times-scales reported appear at odds with the reported time-scales they report in their other work (Ref #17, Bordbar, et al).

The labeling of metabolites in Figure 4A is unclear. How are "high-energy" versus "low-energy" substrates and cofactors defined? For example, why are D-glucose and inorganic phosphate "high-energy substrates"? These definitions do not correspond to the Gibbs Free Energy values (which might be considered the most intuitive definition). Since the authors are using a different definition, it would help to be explicit about the definitions.

It is unclear if some references are appropriate (or the intended ones), such as the comment regarding "flux-concentration duality" when discussing the mapping between the modal matrix and concentration pools, but refers to a manuscript focused on the steady state flux balance problem and LP duality (Ref #36, Warren and Jones, PRL) (Lines 356 ad 635).

Reviewer #2 (Comments for the Author):

The paper by Akbari et al. presents a computational framework for the timescale decomposition of biochemical reaction networks. This framework is designed to extract essential patterns from the complex dynamics of these networks, especially in the context of evolving environments. The paper introduces a computational framework for dissecting the dynamics of biochemical reaction networks, with a focus on timescales, concentration pools, and coherent structures. The authors illustrate its utility through the analysis of kinetic models, highlighting its potential for understanding the dynamics of biological systems and its relevance for applications such as personalized medicine.

Strengths

- The paper introduces an innovative computational approach, Dynamic Mode Analysis (DMA), for dissecting the dynamics of complex biological systems in evolving environments. DMA effectively characterizes the timescale hierarchies of complex reaction networks, providing a system-level understanding at physiologically relevant timescales.
- The paper adeptly identifies the timescales and their associated concentration pools within hypothetical pathways, and it accurately characterizes the transitory regime between consecutive timescales.
- The authors provide a valuable proof of the relationship between time-delayed autocorrelation, covariance, and local Jacobian spectra, which contributes to a deeper understanding of the proposed framework.

Weaknesses

- It would be beneficial for the authors to explore whether similar methods have been developed and provide a comparative analysis, highlighting the strengths and weaknesses of their approach in relation to existing techniques.
- The paper lacks a discussion of the limitations of the DMA approach or suggestions for potential improvements. Addressing these aspects would enhance the paper's completeness and utility.
- The paper could benefit from a more comprehensive discussion of the implications of the results, as much of the text currently appears to be repetitive.
- While Figure 1 is informative, it would be advantageous to include an additional descriptive pipeline of DMA as a central component of Figure 1. This supplementary diagram could elucidate the processing steps of DMA, providing readers with a clearer understanding of how the method operates.

Specific Edits

- In line 256, it is recommended to rephrase the sentence: "The pathway imports Metabolite 1 from and exports Metabolite 4 into the extracellular environment with $x * 1$ and $x * 4$, the respective extracellular concentrations."
- Regarding the citations, in line 55, the authors cite reference No. 11 for "A key component of DMA is an extension of Dynamic Mode Decomposition (DMD)." In line 207, references No. 19 and 20 are cited for "A key component of our computational framework is an extended version of Optimal Dynamic Mode Decomposition [19, 20]". It is confusing.

Code Availability:

- In line 867, it is mentioned that the authors provide code in the Supplementary Information files. However, the absence of the code is noted. It is highly recommended that the authors consider making their code available on a platform such as GitHub. This would greatly assist readers and computational biologists in understanding and implementing the method effectively. The provision of code and its description is invaluable for reproducibility and practical application.

We thank the reviewers for their comments. Our point-by-point response to the reviewer's comments are provided below.

Reviewer #1 (Comments for the Author):

Akbari, Haiman, and Palsson describe a method for calculating subsets (or pools) metabolites according to enzyme kinetics of reaction pathways in the manuscript entitled, 'A data-driven approach for timescale decomposition of biochemical reaction networks'. The Palsson group has published numerous papers on this topic over the past years, including textbooks, so the content has largely been described previously for metabolic and signaling networks. The novel contribution being added here is "Dynamic Model Analysis" for calculating dynamic pools of metabolites that is closely related to autocorrelation analysis. The present manuscript is lucid, descriptively systematic, and well-written. There are a few points that may be considered in revising the manuscript.

The presented approach essentially involves linearization following a perturbation and then application of a sliding window. Based on this, it would appear that the definition of the pools are all dependent on the type(s) of perturbation that is(are) applied. This limits the generalizability of the approach and should probably be discussed.

We have added a paragraph at the end of section Dynamic Mode Analysis in the Methods section of the revised manuscript to address this comment (highlighted on page 23 of the revised manuscript).

It is not clear how the tolerance threshold (11a, 11b, Line 335) is selected. This appears to be a somewhat arbitrary parameter that can have significant impact on the ultimate calculation of the pooling matrix.

Similarly to tolerance thresholds in other model reduction methods, the tolerance threshold in Eq. (11) is a user-defined parameter. We note that the inverse of the dominant eigenvalues in each time window can only be regarded as an approximation of the timescales associated with the concentration pools forming in that time window. Therefore, the purpose of Eq. (11) is to make the definition of timescales more mathematically precise and objective. Granted, the timescales T ascertained from this equation can vary depending on the choice of concentration scales \mathcal{C} . However, once an appropriate concentration scale is selected, the timescales T_1, T_2, T_3, \dots furnished by this equation are expected to capture the timescale hierarchy of the system.

The dynamics for the glycolytic example (particularly for the first 5 pools) seem very slow in terms of what is typically expected for small molecule metabolism (see for example, cited reference #9, Kauffman et al). It is not clear if this is due to the model that was used versus a limitation of the method. For example, a statement is made with regards to consistency with circulation times (Line 159), however the times-scales reported appear at odds with the reported time-scales they report in their other work (Ref #17, Bordbar, et al).

The pseudo-elementary rate constants we used in this study are based on the values estimated for the default glycolysis model in MassPy, as previously reported [1]. Parameter estimation for this glycolysis model were performed based on updated values of steady-state concentrations and equilibrium constants compared to those used in previous studies [2,3]. Notable examples

are the enzymes PGI and PGM, where $k_{PGI} = 1.0 \times 10^5 \text{ hr}^{-1}$ and $k_{PGM} = 1.5 \times 10^4 \text{ hr}^{-1}$ were reported in [2], while the corresponding values in this work were $k_{PGI} = 3.6 \times 10^3 \text{ hr}^{-1}$ and $k_{PGM} = 4.8 \times 10^3 \text{ hr}^{-1}$, which explains the difference in the timescales of the upper/lower glycolysis pools between our results and previous works. However, the formation of pools and their timescales follow a similar pattern qualitatively. With regards to the circulation timescale, DMA identified local complex eigenvalues in the range $1 \text{ min} \lesssim t \lesssim 10 \text{ hr}$, indicating that the dynamics are oscillatory in this interval, which is what we are referring to in the sentence the reviewer is referring to. We referred to $T = 1 \text{ min}$ as the circulation timescale, which is the same value that Bordbar et al [4] reported. Since the upper limit of the interval is on the same order as the circadian timescale, we referred to it as the circadian period.

The labeling of metabolites in Figure 4A is unclear. How are "high-energy" versus "low-energy" substrates and cofactors defined? For example, why are D-glucose and inorganic phosphate "high-energy substrates"? These definitions do not correspond to the Gibbs Free Energy values (which might be considered the most intuitive definition). Since the authors are using a different definition, it would help to be explicit about the definitions.

Our definition of high- or low-energy substrates and cofactors is based on the standard transformed Gibbs energy of formation of metabolites. In the case of phosphate specifically, depending on the form in which it is imported from the extracellular environment, it may be regarded as a high- or low-energy substrate. As an ion in an aqueous form, phosphate PO_4^{3-} is to be regarded as a low-energy cofactor. We have added a subsection to Methods (highlighted on page 30 of the revised manuscript), titled "Energetics of Glycolysis", to clarify our definition of high- and low-energy substrates/cofactors. We also corrected the labeling of Figure 4A, referring to inorganic phosphate as a low-energy substrate in the revised manuscript.

It is unclear if some references are appropriate (or the intended ones), such as the comment regarding "flux-concentration duality" when discussing the mapping between the modal matrix and concentration pools, but refers to a manuscript focused on the steady state flux balance problem and LP duality (Ref #36, Warren and Jones, PRL) (Lines 356 ad 635).

We believe that the duality of concentration pools and decay modes in our analysis and the duality of steady-state flux and chemical potential in the work of Warren and Jones are similar and related concepts. In our work, concentration pools are linked to the thermodynamic driving force of reactions, so they are a proxy for fluxes (measured in units of concentration) and are naturally represented in a flux space, while decay modes are naturally represented in a concentration space. In the work of Warren and Jones, steady-state fluxes are naturally represented in a flux space, while chemical potentials are a proxy for concentrations and are naturally represented in a concentration space. We believe that this duality can be best captured geometrically by another concept of duality commonly used in Differential Geometry to construct invariant quantities, namely the duality of vectors and co-vectors that naturally live in tangent and co-tangent spaces [5]. Here, we briefly explain the concept of duality in the context of Lagrangian mechanics and discuss its analogous counterpart for reaction systems (i.e. flux-concentration duality) at the end.

We consider the dynamics of an N particle system subject to a conservative external force field F_i . In this system, the concept of duality is trivial when the motion of particles is unconstrained, so

we focus on a case where the particles are constrained to move on a manifold defined by the following constraints

$$g_j(\mathbf{r}_1, \mathbf{r}_2, \dots, \mathbf{r}_N) = 0, \quad j = 1 \dots m,$$

where \mathbf{r}_i is the position vector of particle i . Accordingly, we have $n = 3N - m$ degrees of freedom and the dynamic trajectories move in an n -dimensional space. For this system, the significance of the Lagrangian formulation is in the fact that the equations of motion are the same regardless of the choice of coordinate system in the n -dimensional manifold, so they are invariant under any coordinate transformations [Chapter 1, 7]. We can parametrize the position vectors \mathbf{r}_i with respect to an arbitrary coordinate system q_1, q_2, \dots, q_n

$$\mathbf{r}_i = \mathbf{r}_i(q_1, q_2, \dots, q_n)$$

The equation of motion with respect to this coordinate system is [6]

$$\frac{d}{dt} \left(\frac{\partial L}{\partial \dot{q}_k} \right) - \frac{\partial L}{\partial q_k} = 0, \quad k = 1 \dots n,$$

where $\dot{q}_k = dq_k/dt$ with $L = T - V$ the Lagrangian, T the kinetic energy, and V the potential energy such that $Q_k = -\partial V/\partial q_k$ are the generalized forces associated with \mathbf{F}_i . This system admits an extremum principle, so the equation of motion can be represented as solutions of an optimization problem, where the action functional

$$I[\mathbf{q}(t)] = \int L(\mathbf{q}, \dot{\mathbf{q}}, t) dt$$

is to be minimized with respect to $\mathbf{q}(t)$. The steady state configuration can also be represented as the solution of another optimization problem, where $V(\mathbf{q})$ is to be minimized with respect to \mathbf{q}^{ss} . The state of this system can be specified in the $2n$ -dimensional generalized position space $(\mathbf{q}, \dot{\mathbf{q}})$. As with other optimization problems, this optimization problem has a dual [Chapter 5, 9], where the dual Lagrangian L' can be derived from the primal Lagrangian L through the Legendre transform

$$L'(\mathbf{p}, \dot{\mathbf{p}}, t) = \min_{(\mathbf{q}, \dot{\mathbf{q}})} \dot{\mathbf{p}}^T \mathbf{q} + \mathbf{p}^T \dot{\mathbf{q}} - L(\mathbf{q}, \dot{\mathbf{q}}, t)$$

to represent the equations of motion in the $2n$ -dimensional generalized momentum space $(\mathbf{p}, \dot{\mathbf{p}})$. Central to the Lagrangian formulation of equations of motion is the fact that both L and L' are regarded as fundamental physical quantities measured in units of energy, so they must be coordinate invariant. Accordingly, to achieve a coordinate invariant formulation of the equations of motion in both primal and dual space, $\dot{\mathbf{p}}^T \mathbf{q}$ and $\mathbf{p}^T \dot{\mathbf{q}}$ must also be coordinate invariant quantities measured in units of energy. This can in turn be achieved by regarding these terms as inner products of a vector and co-vector, representing $(\mathbf{q}, \dot{\mathbf{q}})$ with respect to a primal basis and $(\mathbf{p}, \dot{\mathbf{p}})$ with respect to the dual basis associated with tangent and co-tangent spaces of the flow manifold [5]. Thus \mathbf{p} is the dual variable corresponding to $\dot{\mathbf{q}}$ at any point along dynamic trajectories and in particular at the steady state. The same is true for $\dot{\mathbf{p}}$ and \mathbf{q} .

In Flux-Balance Analysis, the maximization of the biomass objective subject to mass-balance constraints to ascertain steady-state fluxes is similar to the minimization of the potential function $V(\mathbf{q})$ to determine equilibrium points in Lagrangian mechanics. Warren and Jones derived the dual for this optimization problem using the Legendre transform of the biomass objective, where the dual objective is the total exchange energy dissipation rate. In this formulation, both biomass

objective (if scaled properly) and the dual objective are also fundamental physical quantities measured in units of energy dissipation rate, which we believe must remain invariant with respect to coordinate transformations. Moreover, in this formulation, chemical potentials and fluxes are dual variables in the same sense that generalized positions and momenta are in Lagrangian mechanics. However, unlike Hamiltonian systems examined in the Lagrangian formulation, the dynamics of dissipative systems cannot always be given as solutions of an optimization problem. Nevertheless, dissipative systems with slow dynamics, like biochemical reaction networks, admit an extremum principle, where the total dissipation functional is to be minimized instead of the action functional [8]. Therefore, chemical potential and fluxes (or concentration pools and decay modes) can be regarded as dual variables at any point along dynamic trajectories and in particular at the steady state. From this point of view, flux-concentration duality in the work of Warren and Jones should be regarded as a special case of a more general principle.

Reviewer #2 (Comments for the Author):

The paper by Akbari et al. presents a computational framework for the timescale decomposition of biochemical reaction networks. This framework is designed to extract essential patterns from the complex dynamics of these networks, especially in the context of evolving environments. The paper introduces a computational framework for dissecting the dynamics of biochemical reaction networks, with a focus on timescales, concentration pools, and coherent structures. The authors illustrate its utility through the analysis of kinetic models, highlighting its potential for understanding the dynamics of biological systems and its relevance for applications such as personalized medicine.

Strengths

- The paper introduces an innovative computational approach, Dynamic Mode Analysis (DMA), for dissecting the dynamics of complex biological systems in evolving environments. DMA effectively characterizes the timescale hierarchies of complex reaction networks, providing a system-level understanding at physiologically relevant timescales.
- The paper adeptly identifies the timescales and their associated concentration pools within hypothetical pathways, and it accurately characterizes the transitory regime between consecutive timescales.
- The authors provide a valuable proof of the relationship between time-delayed autocorrelation, covariance, and local Jacobian spectra, which contributes to a deeper understanding of the proposed framework.

Weaknesses

- It would be beneficial for the authors to explore whether similar methods have been developed and provide a comparative analysis, highlighting the strengths and weaknesses of their approach in relation to existing techniques.

We have added a paragraph to Introduction, discussing various model reductions techniques with their strengths and weaknesses (highlighted on page 2 of the revised manuscript).

- The paper lacks a discussion of the limitations of the DMA approach or suggestions for potential improvements. Addressing these aspects would enhance the paper's completeness and utility.

We have added a paragraph to Discussion, elaborating on the limitations of DMA related to noise sensitivity and offered potential techniques that could be implemented in the future to alleviate the sensitivity of DMA to noise in the data (first highlighted paragraph of Discussion on page 11 of the revised manuscript).

- The paper could benefit from a more comprehensive discussion of the implications of the results, as much of the text currently appears to be repetitive.

We have added a paragraph to Discussion, talking about broader implications of DMA (second highlighted paragraph of Discussion on page 11 of the revised manuscript).

- While Figure 1 is informative, it would be advantageous to include an additional descriptive pipeline of DMA as a central component of Figure 1. This supplementary diagram could elucidate the processing steps of DMA, providing readers with a clearer understanding of how the method operates.

The workflow of Dynamic Mode Analysis is illustrated in Fig. 1. However, this figure does not show the steps of the extension of Optimal Dynamic Mode Decomposition, which we applied to determine the dominant modes from time-series data. We have added a figure to the SI document (Fig. S4), illustrating how dominant modes are determined from time-series data in our extended version of Optimal Dynamic Mode Decomposition, referring to it in the main manuscript (highlighted on page 20 of the revised manuscript).

Specific Edits

- In line 256, it is recommended to rephrase the sentence: "The pathway imports Metabolite 1 from and exports Metabolite 4 into the extracellular environment with $x*1$ and $x*4$, the respective extracellular concentrations."

We have rephrased this sentence in the revised manuscript (highlighted on page 12 of the revised manuscript).

- Regarding the citations, in line 55, the authors cite reference No. 11 for "A key component of DMA is an extension of Dynamic Mode Decomposition (DMD)." In line 207, references No. 19 and 20 are cited for "A key component of our computational framework is an extended version of Optimal Dynamic Mode Decomposition [19, 20]". It is confusing.

DMA could be regarded as an extension of the original Dynamic Mode Decomposition discussed in reference [11] or its subsequent extensions, such as the Optimal Dynamic Mode Decomposition in reference [19], which is also discussed in more detail in the sparsity promoting version of it in reference [20]. To avoid confusion, we cited all three references in both Introduction and Discussion of the revised manuscript.

Code Availability:

- In line 867, it is mentioned that the authors provide code in the Supplementary Information files. However, the absence of the code is noted. It is highly recommended that the authors consider making their code available on a platform such as GitHub. This would greatly assist readers and computational biologists in understanding and implementing the method effectively.

The provision of code and its description is invaluable for reproducibility and practical application.

We provided a link to a GitHub repository containing the codes we used to perform the analyses and generate the results presented in this paper and modified the Code Availability section accordingly.

References

- [1] Haiman, Z.B., Zielinski, D.C., Koike, Y., Yurkovich, J.T., Palsson, B.O.. Masspy: Building, simulating, and visualizing dynamic biological models in 859 python using mass action kinetics. *PLoS Comput Biol* 2021;17(1):e1008208
- [2] Jamshidi, N. and Palsson, B.Ø., 2008. Formulating genome-scale kinetic models in the post-genome era. *Molecular systems biology*, 4(1), p.171.
- [3] Kauffman, K.J., Pajerowski, J.D., Jamshidi, N., Palsson, B.O. and Edwards, J.S., 2002. Description and analysis of metabolic connectivity and dynamics in the human red blood cell. *Biophysical Journal*, 83(2), pp.646-662.
- [4] Bordbar, A., McCloskey, D., Zielinski, D.C., Sonnenschein, N., Jamshidi, N. and Palsson, B.O., 2015. Personalized whole-cell kinetic models of metabolism for discovery in genomics and pharmacodynamics. *Cell systems*, 1(4), pp.283-292.
- [5] Ivancevic, V.G. and Ivancevic, T.T., 2007. *Applied differential geometry: a modern introduction*. World Scientific.
- [6] Arnol'd, V.I., 2013. *Mathematical methods of classical mechanics (Vol. 60)*. Springer Science & Business Media.
- [7] Gelfand, I.M. and Fomin, S.V., 2000. *Calculus of Variations*, (Translated and edited by Silverman, RA).
- [8] Herivel, J.W., 1953, January. A general variational principle for dissipative systems. In *Proceedings of the Royal Irish Academy. Section A: Mathematical and Physical Sciences (Vol. 56, pp. 37-44)*. Royal Irish Academy.
- [9] Boyd, S.P. and Vandenberghe, L., 2004. *Convex optimization*. Cambridge university press.

Re: mSystems01001-23R1 (A data-driven approach for timescale decomposition of biochemical reaction networks)

Dear Prof. Bernhard O Palsson:

Thank you for providing a revised version of your manuscript. All reviewer comments were addressed carefully, and I am now happy to accept the revised manuscript for publication.

Your manuscript has been accepted, and I am forwarding it to the ASM production staff for publication. Your paper will first be checked to make sure all elements meet the technical requirements. ASM staff will contact you if anything needs to be revised before copyediting and production can begin. Otherwise, you will be notified when your proofs are ready to be viewed.

Featured Image Submissions: If you would like to submit a potential Featured Image, please email a file and a short legend to mSystems@asmusa.org. Please note that we can only consider images that (i) the authors created or own and (ii) have not been previously published. By submitting, you agree that the image can be used under the same terms as the published article. File requirements: square dimensions (4" x 4"), 300 dpi resolution, RGB colorspace, TIF file format.

Sincerely,
Lennart Schada von Borzyskowski
Editor